# Inhibitor-3 inhibits Protein Phosphatase 1 via a metal binding dynamic protein–protein interaction

Gautam Srivastava [1], Meng S. Choy[1], Nicolas Bolik-Coulon[2], Rebecca Page[3] & Wolfgang Peti [1] ✉

To achieve substrate specificity, protein phosphate 1 (PP1) forms holoenzymes with hundreds of regulatory and inhibitory proteins. Inhibitor-3 (I3) is an ancient inhibitor of PP1 with putative roles in PP1 maturation and the regulation of PP1 activity. Here, we show that I3 residues 27–68 are necessary and sufficient for PP1 binding and inhibition. In addition to a canonical RVxF motif, which is shared by nearly all PP1 regulators and inhibitors, and a non-canonical SILK motif, I3 also binds PP1 via multiple basic residues that bind directly in the PP1 acidic substrate binding groove, an interaction that provides a blueprint for how substrates bind this groove for dephosphorylation. Unexpectedly, this interaction positions a CCC (cys-cys-cys) motif to bind directly across the PP1 active site. Using biophysical and inhibition assays, we show that the I3 CCC motif binds and inhibits PP1 in an unexpected dynamic, fuzzy manner, via transient engagement of the PP1 active site metals. Together, these data not only provide fundamental insights into the mechanisms by which IDP protein regulators of PP1 achieve inhibition, but also shows that fuzzy interactions between IDPs and their folded binding partners, in addition to enhancing binding affinity, can also directly regulate enzyme activity.

Protein Phosphatase 1 (PP1; 37.5 kDa) is the most widely expressed and abundant serine/threonine phosphatase. It is also exceptionally well-conserved, from fungi to humans, in both sequence and function[1]. Dephosphorylation events catalyzed by PP1 regulate multiple processes, including cell-cycle progression, protein synthesis, muscle contraction, carbohydrate metabolism, transcription and neuronal signaling, among others. All phosphoprotein phosphatases (PPPs), including PP1, require two metal ions at their active site (M1 and M2), which activate a water molecule to hydrolyze the phosphate group from substrates[2]. Although the intrinsic specificity of PP1 is low, PP1 achieves exquisite specificity by interacting with >200 known regulatory proteins. These regulators target PP1 to distinct cellular compartments and direct its activity toward specific substrates[3]. The majority of these regulators are intrinsically disordered proteins (IDPs)

and bind PP1 via a short RVxF short linear motif (SLiM); however, additional motifs beyond the RVxF SLiM are often critical for cellular function and allow for a molecular distinction between these regulators[4,5].

Two evolutionarily ancient PP1 regulators, expressed in all lineages of eukaryotes, are suppressor-of-Dis2-number 2 (SDS22; PPP1R7)[6] and Inhibitor-3 (126 aa; I3; PPP1R11 also referred to as YPI1, HCGV, IPP3, or TCTEX5; Fig. 1A)[7,8]. Both SDS22 and I3 are essential genes, as their individual deletion in yeast is lethal[9,10]. Furthermore, conditional loss of either protein alters the cellular localization of PP1 and results in growth arrest, demonstrating that these genes are essential for PP1 function[11]. As is typical for most PP1 regulators, SDS22 and I3 form heterodimeric complexes with PP1 (SDS22:PP1 and I3:PP1)[9,12]. However, they also form a heterotrimeric complex (SDS22:I3:PP1)[13]. Despite their

[1]Department of Molecular Biology and Biophysics, University of Connecticut Health Center, Farmington, CT, USA. [2]Department of Chemistry and Biochemistry, University of Arizona, Tucson, AZ, USA. [3]Department of Cell Biology, University of Connecticut Health Center, Farmington, CT, USA. ✉e-mail: peti@uchc.edu

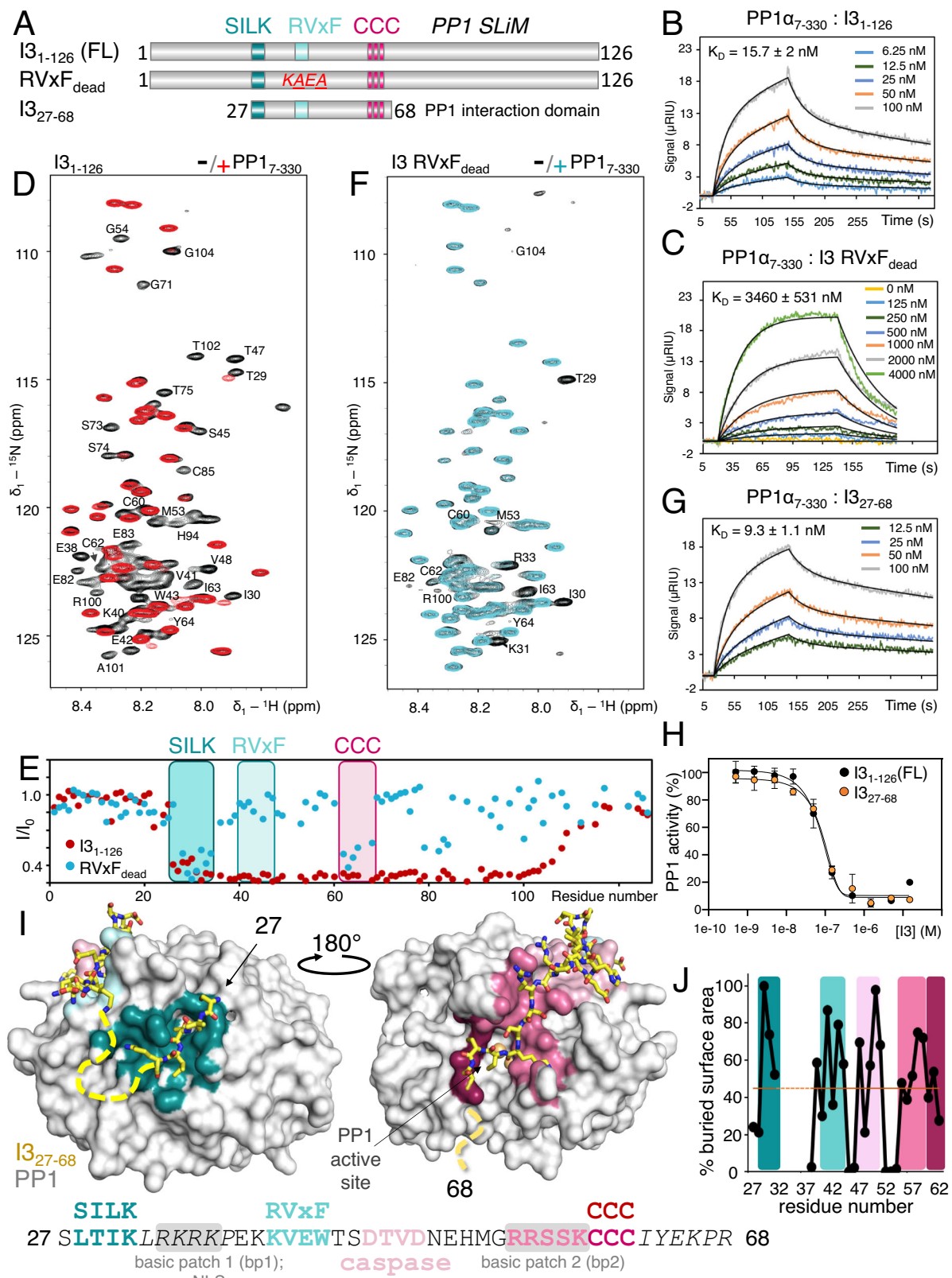

**Fig. 1 | I3 interaction with PP1. A** Domain structure of I3. PP1-specific SLiMs (SILK [teal], RVxF [turquoise] and CCC [dark pink]) are indicated. Additional constructs relevant to this figure are also shown. **B** SPR sensorgram for I3$_{1-126}$ and PP1α$_{7-330}$. **C** SPR sensorgram for I3$_{1-126}$ RVxF$_{dead}$ and PP1α$_{7-330}$. **D** 2D [$^{1}$H,$^{15}$N] HSQC spectrum of $^{15}$N-labeled I3$_{1-126}$ alone (black) and in complex with PP1α$_{7-330}$ (red). **E** Peak intensity loss vs I3 protein sequence plot; SLiMs indicated with the same colors shown in **A**. **F** 2D [$^{1}$H,$^{15}$N] HSQC spectrum of $^{15}$N-labeled I3$_{1-126}$ RVxF$_{dead}$ alone (black) and in complex with PP1α$_{7-330}$ (cyan). **G** SPR sensorgram for the minimal PP1

interaction domain of I3, I3$_{27-68}$, and PP1α$_{7-330}$. **H** IC$_{50}$ curves for PP1 inhibition by I3$_{1-126}$ and I3$_{27-68}$. IC$_{50}$ values reported in Table 1. Data are presented as mean values ± SD, $n = 4$ technical replicates. **I** Crystal structure of the I3$_{27-68}$:PP1α$_{7-300}$ complex with I3 shown as yellow sticks and PP1 shown as a surface; colors correspond to the binding pockets of the I3 motifs highlighted in the I3 sequence (below). I3 residues not modeled due to a lack of clear electron density indicated by dashed lines (also shown in *italics* in the sequence). **J** Plot of the I3 solvent accessible surface area buried per residue when I3 is bound to PP1.

**Table 1 | I3:PP1 binding and inhibition**

| PP1 and I3 variants | Binding | | | Inhibition | |
|---|---|---|---|---|---|
| **PP1$_{7-330}$ vs I3 variants** | **K$_D$ (nM)** | **Fit** | ***n*** | **IC$_{50}$ (nM)** | ***n*** |
| **I3$_{1-126}$ (FL, full-length)** | | | | | |
| I3$_{1-126}$ | 15.7 ± 2 | 2 | 3 | 71 ± 6 | 3 |
| SILK$_{dead}$: I3$_{1-126}$ T29A/I30A/K31A | 13.1 ± 2.8 | 2 | 3 | nd | nd |
| RVxF$_{dead}$: I3$_{1-126}$ $^{40}$KAEA$^{43}$ | 3460 ± 531 | 1 | 4 | nd | nd |
| SSS: I3$_{1-126}$ C60S/C61S/C62S | 300 ± 23 | 1 | 3 | nd | nd |
| **I3$_{27-68}$ (I3 PP1 interaction domain)** | | | | | |
| pH = 8.0 | 9.3 ± 1.1 | 2 | 4 | 75 ± 4 | 8 |
| pH = 7.5 | 9 ± 0.6 | 2 | 3 | nd | nd |
| pH = 7.0 | 16.1 ± 1.1 | 2 | 3 | nd | nd |
| pH = 6.5 | 17 ± 0.8 | 2 | 3 | nd | nd |
| SILK$_{dead}$: I3$_{27-68}$ T29A/I30A/K31A | 16.8 ± 2.8 | 2 | 3 | nd | nd |
| SCC: I3$_{27-68}$ C60S | 21.7 ± 3.1 | 2 | 5 | 243 ± 13 | 4 |
| CSC: I3$_{27-68}$ C61S | 18.8 ± 0.9 | 2 | 3 | 158 ± 13 | 4 |
| CCS: I3$_{27-68}$ C62S | 20.6 ± 5.1 | 2 | 3 | 171 ± 7 | 4 |
| CSS: I3$_{27-68}$ C61S/C62S | 21.2 ± 0.6 | 2 | 2 | 153 ± 10 | 4 |
| SCS: I3$_{27-68}$ C60S/C62S | 54 ± 9 | 2 | 3 | 179 ± 15 | 4 |
| SSC: I3$_{27-68}$ C60S/C61S | 86.8 ± 3.8 | 1 | 4 | 349 ± 19 | 4 |
| SSS: I3$_{27-68}$ C60S/C61S/C62S | 412 ± 6 | 1 | 3 | 380 ± 31 | 4 |
| IY: I3$_{27-68}$ I63A/Y64A | 15.6 ± 2.4 | 2 | 3 | nd | nd |
| **I3$_{27-68}$ deletion variants** | | | | | |
| SILK$_{deletion}$: I3$_{38-68}$ (Δ27-37) | 15.2 ± 2.5 | 2 | 3 | nd | nd |
| CCC$_{deletion}$: I3$_{27-59}$ (Δ60-68) | 376 ± 47 | 1 | 3 | 422 ± 113 | 4 |
| **I3$_{79-91}$** | | | | | |
| I3$_{79-91}$ | No binding | | 3 | nd | nd |
| **I3$_{27-68}$ vs. different PP1 variants** | | | | | |
| PP1$_{7-330}$ C273S | 16.1 ± 0.7 | 2 | 3 | 85 ± 3 | 8 |
| PP1$_{7-330}$ H66K | 5.9 ± 0.6 | 2 | 2 | nd | nd |
| PP1$_{7-330}$ H248N | 6.6 ± 1.7 | 2 | 2 | nd | nd |

Bold used to highlight different I3 constructs and PP1 variants.

importance for promoting proper PP1 function, when either or both regulators bind PP1, the activity of PP1 is potently inhibited[9,14]. How these proteins simultaneously promote and inhibit PP1 function are key questions in the field.

Recently, we showed that SDS22, a folded protein consisting of mostly leucine-rich repeats (one of the few regulators that lack an RVxF motif)[14–16], selectively binds a conformation of PP1 which lacks its M1 metal. Because both metals are essential for PP1 activity, SDS22-bound PP1 is inactive. Thus, these data showed that SDS22 serves as a cellular PP1 'storage' protein, maintaining PP1 in an inactive state until needed for holoenzyme formation with other PP1-specific regulators. Compared to SDS22, a molecular understanding of how I3 both inhibits and promotes PP1 function is still lacking despite its discovery more than 25 years ago[9,11,17].

Like most PP1 regulators, I3 is predicted to be an IDP, is heat stable and binds PP1 using a PP1 RVxF SLiM ($^{40}$K**V**E**W**$^{43}$; Fig. 1A)[18]. The hydrophobic residues V and F are the key RVxF binding elements, with R/K and F/W identified as common substitutions in biochemically confirmed RVxF motifs[19]. Typically, the 'x' residue in the RVxF motif is an S or T and, when phosphorylated, significantly weakens PP1 binding[20]. This weakening is also achieved by mutating the 'x' residue to a phosphomimetic D/E[21]. Thus, the 'E' at the 'x' position of the RVxF motif is atypical and, furthermore, makes the phosphorylation-induced disruption of RVxF binding impossible for I3. In addition to binding PP1, I3 also potently inhibits PP1 activity[7], a function that requires residues C-terminal to the I3 RVxF motif[18]. However, how, at a molecular level, I3 inhibits PP1 and if and/or how this mechanism is distinct from that

used by SDS22, is still poorly understood. Finally, while multiple triple PP1 holoenzymes have been identified, most are comprised of regulators that *compete* for the PP1 RVxF binding pocket[22–24]. This is different for the SDS22:I3:PP1 triple complex, as only I3 contains an RVxF motif. Thus, the extent to which the interaction sites of SDS22 and I3 on PP1 overlap is currently unknown.

Here, we combined molecular and cell biology studies of the I3:PP1 complex to define how I3 binds and inhibits PP1. We show that I3 and PP1 form a stable PP1 complex. I3 binds PP1 not only via an RVxF motif but also a SILK motif, two basic-rich patches and a previously unidentified CCC motif. Unlike SDS22, which selectively binds an inactive form of PP1 (only a single metal at the active site)[14], the I3:PP1 complex contains two metals at the PP1 active site and thus should be capable of dephosphorylating substrates. However, I3 inhibits PP1. Our structure shows that the I3 CCC motif binds across the active site. Additional experiments revealed that these cysteines dynamically engage the M2 metal in a transient, dynamic/'fuzzy' manner[25,26], as the deletion of one or even two of the cysteine residues, independent of their position within the CCC motif, still allows for efficient PP1 inhibition. Finally, the structures of the SDS22:PP1 and I3:PP1 complexes also show that the interaction surfaces of SDS22 and I3 on PP1 do not overlap, explaining why they can bind PP1 simultaneously. Together, this study provides insights into the mechanism by which I3 inhibits PP1, furthers our understanding of the physiological regulation of PP1 and highlights an inhibitory interaction via a CCC motif that might be shared between all phosphoprotein phosphatase family members.

## Results

### I3 uses residues beyond its RVxF SLiM to bind PP1
To identify the residues in I3 that mediate PP1 binding, we used mutagenesis and surface plasmon resonance (SPR). Full-length I3 binds PP1α$_{7-330}$ tightly, with a K$_D$ of 15.7 ± 2 nM (Fig. 1B; Table 1 and S1). However, while most PP1 regulators bind PP1 with fast on- and off-rates, the SPR sensorgram shows that I3 binds PP1 comparatively slowly and dissociates even more slowly, with biphasic kinetics defined by fast and slow dissociation events. These equilibrium and kinetic data suggest that residues outside the I3 RVxF motif ($^{40}$KVEW$^{43}$) contribute to PP1 binding and that they might engage PP1 via mechanisms distinct from canonical PP1 regulators.

To test if residues outside the RVxF motif are important for binding, we measured the binding affinity of the I3$_{RAXA}$ variant (I3 $^{40}$KVEW$^{43}$ mutated to $^{40}$KAEA$^{43}$; RVxF$_{dead}$ variant; Fig. 1A) to PP1α. As expected, while the measured affinity was significantly lower, (K$_D$, 3460 ± 531 nM; Table 1 and S1; Fig. 1C), there was still measurable binding, confirming that residues outside the RVxF motif facilitate I3:PP1 complex formation and thus likely contribute to its biological function. I3 is a heat stable protein and the 2D [$^1$H,$^{15}$N] heteronuclear single quantum correlation (HSQC) spectrum of $^{15}$N-labeled I3 exhibits all hallmarks of an IDP, including narrow chemical shift dispersion in the $^1$H dimension due to the lack of hydrogen bonds in secondary structure elements (Fig. 1D). Upon completion of the sequence-specific backbone assignment of I3 (Figs. S1A, S1B), a chemical shift index (CSI/SSP) analysis, using Cα and Cβ chemical shifts, showed that I3 has only small preferred α-helical structure for I3 residues 15–30 (~25%) and 45–55 (~20%) as well as a preference for an extended conformation around the predicted RVxF site (typical for many PP1 regulators in their free form[4,27–32]) and the C-terminal residues (Figs. S1C, S1D). However, these secondary structure preferences are minimal compared to those identified for Inhibitor-2[28,33] indicating a likely different mode of interaction with PP1.

To define the I3 PP1 interaction domain, we measured a 2D [$^1$H,$^{15}$N] HSQC spectrum of $^{15}$N-labeled I3 bound to PP1α$_{7-330}$. In this experiment, the cross-peaks corresponding to residues that bind PP1 disappear due to the increased molecular weight of the I3:PP1 complex (~50 kDa), while the peaks corresponding to residues that do not bind

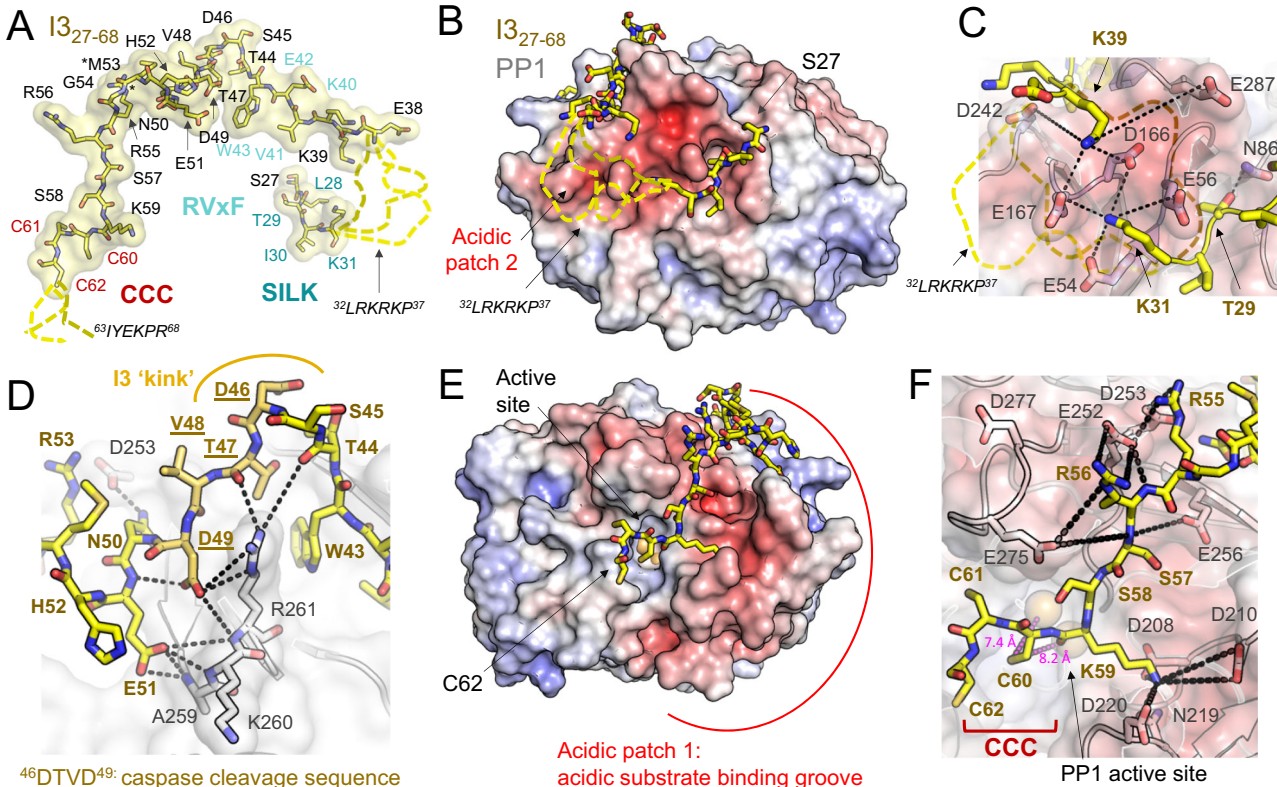

**Fig. 2 | The I3:PP1 holoenzyme. A** The structure of I3 when bound to PP1; residues shown as sticks and labeled. PP1 motif residue labels are colored as in (1A). I3 residues not modeled due to a lack of electron density indicated by dashed lines in yellow hues. **B** The I3:PP1 complex with I3 shown as yellow sticks and PP1 shown as a surface colored according to electrostatic potential. The location of acidic patch 2 is indicated. I3 residues not modeled due to a lack of electron density indicated by dashed lines in yellow hues. **C** Close up of PP1 acidic patch 2. PP1 acidic residues shown as sticks and labeled (gray) with I3 residues shown as sticks (yellow). Electrostatic/polar interactions indicated by black dotted lines. Dynamic I3 residues[31–37] indicated by dashed lines in yellow hues. **D** The I3 'kink'. PP1 shown as a transparent gray surface with I3 shown as sticks in yellow; the I3 caspase cleavage sequence, [46]DTVD[49], is colored slightly darker yellow for clarity. Electrostatic/polar interactions that stabilize the kink and DTVD residues shown as dashed black lines. **E** Same as **B** with the I3:PP1 complex rotated by 180° to view the C-terminal portion of I3 bound to PP1. PP1 acidic patch 1, also known as the PP1 acidic substrate binding groove, is labeled. **F** Close-up of PP1 acidic patch 1. PP1 acidic residues shown as sticks and labeled (gray) with I3 residues shown as sticks (yellow). Electrostatic/polar interactions indicated by dashed black lines. Distances between the Cys60 sulfur atom and the PP1 M1, M2 metals indicated by magenta dotted lines with the distances labeled.

remain visible[29,30,34,35]. An overlay of the free and PP1-bound spectra showed that I3 interacts extensively with PP1, as many I3 peaks disappeared upon 1:1 complex formation (Fig. 1D, E), consistent with the high binding affinity determined by SPR. Next, we repeated the NMR experiment using [15]N-labeled I3 RVxF$_{dead}$. The PP1-bound I3 RVxF$_{dead}$ 2D [[1]H,[15]N] HSQC spectrum showed many fewer cross peaks with reduced intensities (Fig. 1E, F). Nevertheless, two I3 regions (I3 residues 27–31 and 60–68) likely interact with PP1, even in the absence of a functional RVxF motif. Furthermore, while I3 residues 70–106 showed reduced intensities in the wt-I3 PP1 spectrum, they had full intensity in the I3 RVxF$_{dead}$ PP1 interaction spectrum. This suggests that residues I3$_{70–106}$, an I3 region that is highly enriched in charged residues, likely interacts with PP1 using dynamic charge:charge ('fuzzy'[26,36,37]) interactions that are only possible when the RVxF motif is functional and bound (i.e. under tight binding conditions); 'fuzzy' interactions driven by electrostatics have been observed in other PP1 and PPP regulator/substrate interactions[32,38,39]. A peptide of I3 residues 79–91, which includes the majority of the charged residues, does not bind PP1 (Table 1 and S1). To confirm that I3 residues 27–68 are necessary and sufficient for PP1 binding and inhibition, we generated I3$_{27–68}$ and tested its affinity for PP1 using SPR. I3$_{27–68}$ binds PP1 with a K$_D$ of 9.3 ± 1.1 nM (Table 1 and S1; Fig. 1G), essentially identical to that observed for full-length I3, confirming that I3 residues 27–68 constitute the full I3 PP1 interaction domain (I3 binding is also independent of pH; Table 1 and S1, Fig. S2). Consistent with these results, PP1

IC$_{50}$ inhibition assays show that I3 residues 27–68 inhibit PP1 to the same extent as full-length I3 (IC$_{50}$ values: I3$_{1-126}$, 71 ± 6 nM, I3$_{27-68}$, 75 ± 4 nM; Fig. 1H).

### I3 binds the SILK motif binding pocket, the RVxF motif binding pocket and the PP1 acidic substrate binding groove, positioning the I3 CCC motif to engage the PP1 active site

To understand how I3 binds and inhibits PP1, we co-expressed I3$_{27–68}$ and PP1α$_{7–300}$ in Expi293F cells and determined the crystal structure of the I3$_{27–68}$:PP1α$_{7–300}$ complex (Fig. 1I; 2.0 Å resolution; hereafter referred to as the I3:PP1 complex; see Table S2 for data collection and refinement statistics). Two metals are present in the PP1 active site (this is the only structure of mammalian expressed PP1 in which both metals are present at the active site). We also determined the crystal structure of the reconstituted I3:PP1 complex, where both proteins were expressed separately in *E. coli* and co-purified (2.5 Å resolution; Table S2). The structures are identical (RMSD = 0.24 and 0.36 Å² for PP1 and I3, respectively) and thus all further discussion is focused on the mammalian (Expi293F) produced I3:PP1 complex.

As anticipated from the NMR data, the interaction between I3 and PP1 is extensive, burying ~3500 Å² of solvent accessible surface area (Fig. 1J), with strong electron density observed for I3 residues 27–31 and 38–62 (Fig. 2A). While the I3 RVxF motif ([40]KVEW[43]) binds in the PP1 RVxF binding pocket as expected, the remainder of the interactions were previously unknown. First, I3 residues [27]SL**TIK**[31] bind the PP1 S**ILK**

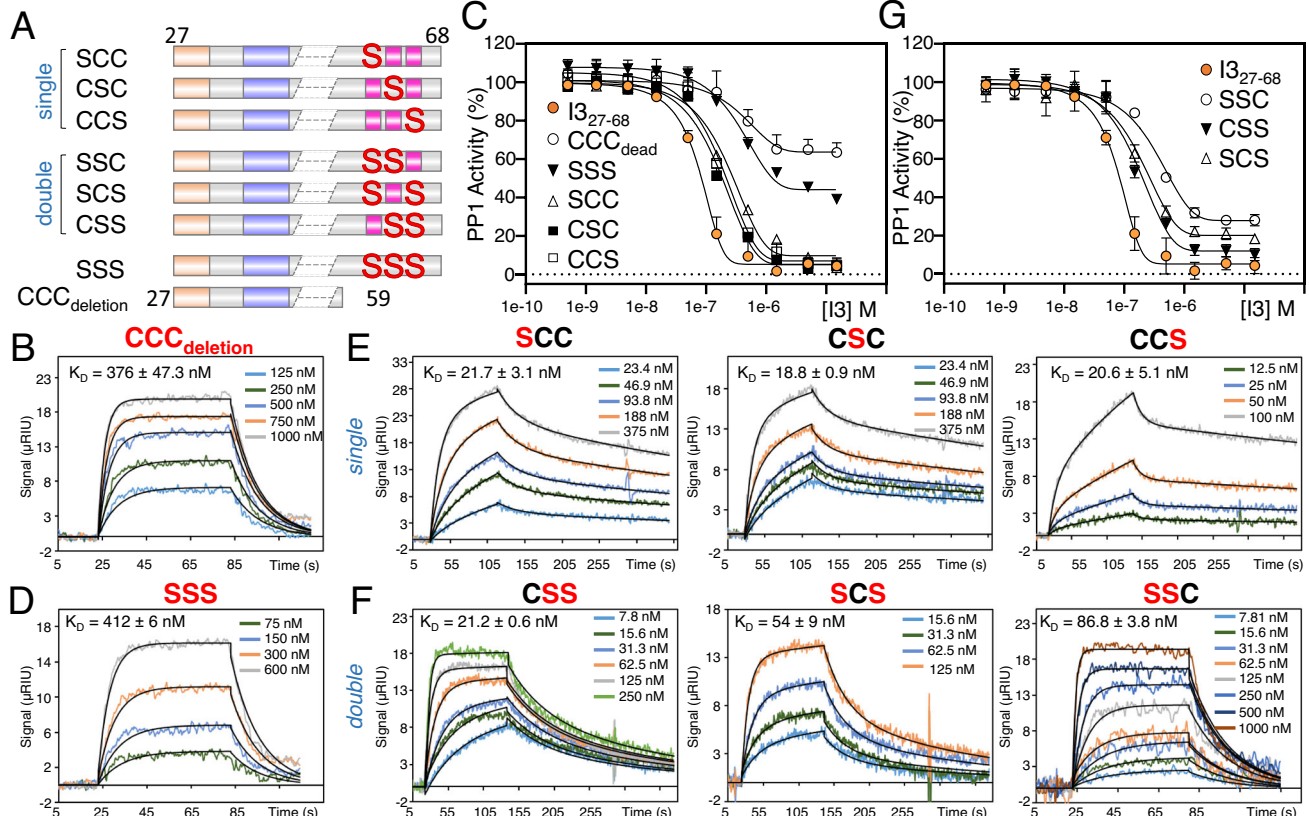

**Fig. 3 | The fuzzy I3 CCC motif. A** Domain structure of I3 constructs for studying the fuzzy I3 CCC motif. **B** SPR sensorgram between the I3 CCC$_{deletion}$ variant and PP1$_{7-330}$. **C** PP1 IC$_{50}$ inhibition for I3$_{27-68}$, I3$_{27-68}$ CCC$_{deletion}$, I3$_{27-68}$ SSS and I3$_{27-68}$ single C→S (SCC, CSC, CCS) mutants. Data are presented as mean values ± SD, $n = 4$ technical replicates. **D** SPR sensorgram between I3$_{27-68}$ SSS and PP1$_{7-330}$. **E** SPR

sensorgrams between I3$_{27-68}$ single C→S (SCC, CSC, CCS) mutants and PP1$_{7-330}$. **F** SPR sensorgrams between I3$_{27-68}$ double CC→SS (SSC, SCS, CSS) mutants and PP1$_{7-330}$. **G** PP1 IC$_{50}$ inhibition for I3$_{27-68}$ and I3$_{27-68}$ double CC→SS (SSC, SCS, CSS) mutants. IC$_{50}$ values reported in Table 1. Data are presented as mean values ± SD, $n = 4$ technical replicates.

binding pocket, but do so using a non-canonical SILK SLiM sequence (Figs. 2B, S3). Namely, Thr29$_{I3}$ binds the pocket commonly occupied by an Ile residue (SILK). The short threonine sidechain does not extend into the deep hydrophobic 'I' pocket, but instead forms a hydrogen bond with PP1 residue Asn86 (Thr29$_{I3}$ hydroxyl – Asn86$_{PP1}$ amine, Fig. 2C). This positions the noncanonical residue Leu28$_{I3}$ to bind the 'S' pocket, and canonical residues Ile30$_{I3}$ and Lys31$_{I3}$ to occupy the 'LK' pockets. This SILK:PP1 interaction allows for a broader definition of experimentally verified SILK motifs.

The I3 residues that connect the SILK and RVxF binding motifs, [32]LRKRKP[37] (bp1: basic patch 1, Fig. 1I; these residues also function as the I3 nuclear localization signal, NLS[40]), lack electron density, suggesting they do not adopt a single conformation (Fig. 2A, B). PP1 residues that define the solvent accessible surface between the SILK and RVxF binding pockets are largely acidic, i.e., electrostatically complementary (acidic patch 2: Glu54$_{PP1}$, Glu56$_{PP1}$, Asp166$_{PP1}$, Glu167$_{PP1}$, Asp242$_{PP1}$, Glu287$_{PP1}$; Fig. 2C). Together, these observations suggest that the I3 bp1 residues form dynamic, 'fuzzy', charge-charge interactions with the acidic PP1 residues.

Following the canonical RVxF interaction, I3 residues [44]TSDT[47] make a sharp turn toward the PP1 active site, positioning the I3 caspase cleavage sequence, [46]DTVD[49] (Fig. 2D)[41], to bridge the PP1 RVxF and the acidic substrate binding groove (Fig. 2E). These highly conserved I3 residues are stabilized by polar and electrostatic interactions between the I3 [49]DNE[51] sequence and PP1 (Asp49$_{I3}$-Arg261$_{PP1}$; Asn50$_{I3}$-Asp253$_{PP1}$; Asn50$_{I3}$-Phe257$_{PP1}$ backbone amide; Glu51$_{I3}$-Ala259$_{PP1}$/Lys260$_{PP1}$ backbone amides; Fig. 2D). The position of Glu51$_{I3}$ is further stabilized by His52$_{I3}$ via π-stacking while the sidechain of Met53$_{I3}$ turns backward toward Asn50$_{I3}$. Together these well-ordered residues lock I3 into a

rigid, unique position on the PP1 surface. Superposition of the I3:PP1 complex with a Caspase-3:peptide inhibitor complex[42] via the DTVD cleavage sequence results in extensive clashing between PP1 and Caspase-3 (Fig. S4), suggesting that PP1 likely inhibits Caspase-3 mediated I3 cleavage.

The next I3 residues, [54]GRRSS[58], bind the PP1 acidic substrate binding groove (acidic patch 1, Fig. 2E) with Arg55$_{I3}$ and Arg56$_{I3}$ forming electrostatic interactions with Glu252$_{PP1}$, Asp253$_{PP1}$, Glu275$_{PP1}$ and Glu256$_{PP1}$ (Fig. 2F). These electrostatic interactions are likely somewhat more dynamic as the B-factors of I3 residues [54]GRRSS[58] are higher, especially for the arginine sidechains (Fig. 2E), than the I3 RVxF and Caspase-3 motif residues. Consistent with this, the adjacent PP1 β12-β13 loop ([273]CGEFD[277]) is also not well-ordered, as it changes conformation in order accommodate I3 in the acidic groove. The next residue with strong sidechain electron density is Lys59$_{I3}$, which binds a deep, acidic pocket on PP1 (Fig. 2F; Asp208$_{PP1}$, Asp210$_{PP1}$, Asn219$_{PP1}$, Asp220$_{PP1}$). Thus, while I3 residues [54]GRR[56], especially the charged sidechains, are dynamic, Lys59$_{I3}$ is well-ordered and tightly associated with PP1. This interaction is critical as it positions I3 residues [60]CCC[62], hereafter referred to as the CCC motif, to bind directly over the PP1 active site (Fig. 2F). No electron density for I3 was observed beyond I3 residue Cys62.

**The I3 CCC motif is critical for PP1 binding and inhibition**
To determine the individual contributions of the distinct I3 SLiM motifs for PP1 binding and inhibition, we used SPR and inhibition assays, respectively. We quantified the contribution of the I3 SILK motif. I3 variants that lack this motif (SILK$_{dead}$: I3$_{1-126}$[29]AAA[31], I3$_{27-68}$[29]AAA[31], SILK$_{deletion}$: I3$_{38-68}$) bind PP1 with affinities nearly

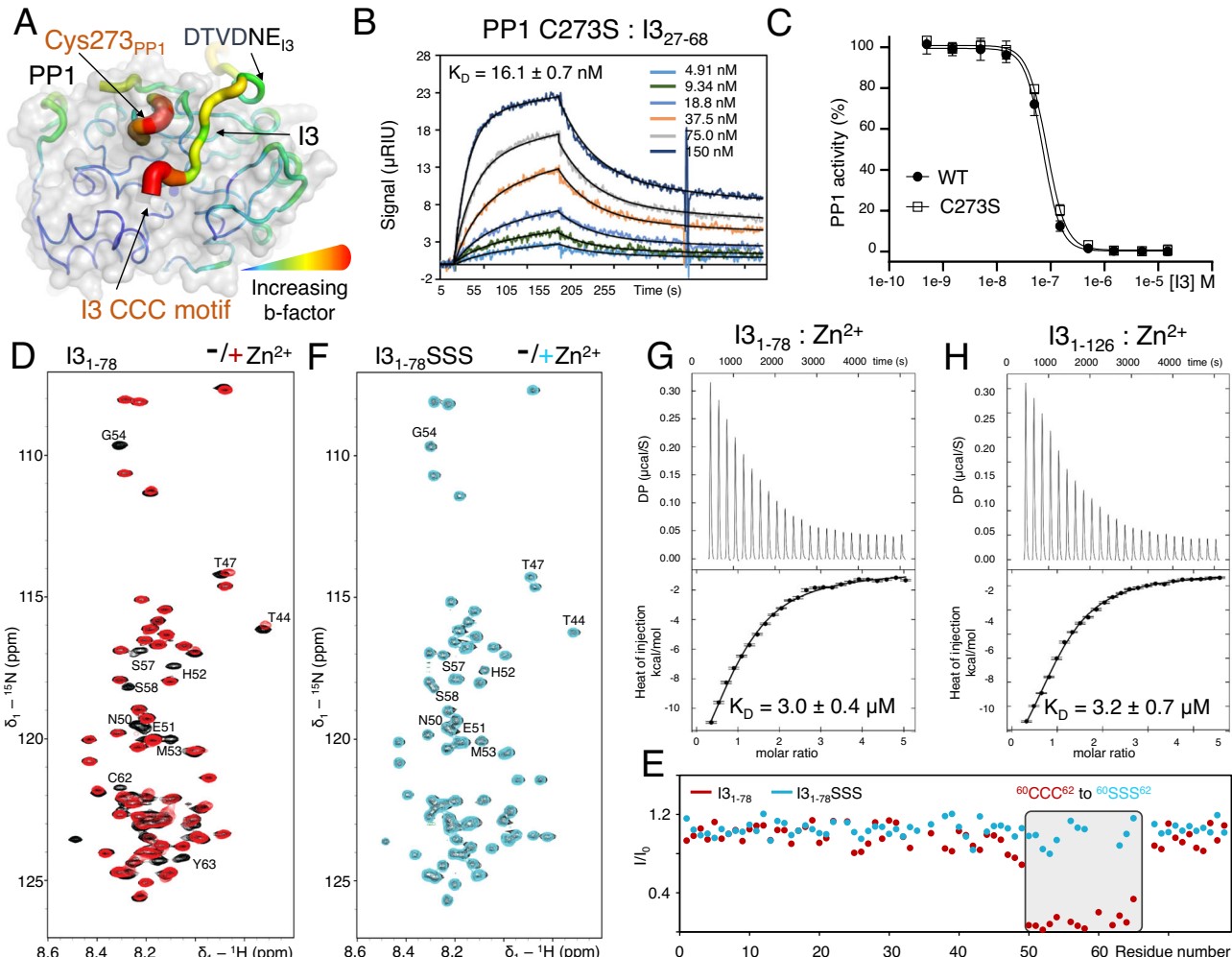

**Fig. 4 | The I3 CCC motif – PP1 active site interaction. A** Structure of the I3:PP1 complex. PP1 and I3 are shown as a pymol sausage plot to reflect relative B factor values (thin blue lines, lowest B factors; fat red lines, highest B factors). PP1 transparent surface also shown in gray. **B** SPR sensorgram between I3$_{27-68}$ and PP1α$_{7-330}$ C273S. **C** IC$_{50}$ data for I3$_{27-68}$ with PP1α$_{7-330}$ C273S and WT. Data are presented as mean values ± SD, $n = 8$ technical replicates. **D** 2D [$^1$H,$^{15}$N] HSQC spectrum of $^{15}$N-labeled I3$_{1-78}$ alone (black) and after the addition of a 4-fold molar excess of Zn$_2$SO$_4$ (red). **E** Peak intensity loss vs I3 protein sequence plot upon addition of

Zn$_2$SO$_4$ for I3$_{1-78}$ (red) and I3$_{1-78}$SSS (cyan). Gray highlight indicates residues that experience a loss of intensity for only WT I3$_{1-78}$ in the presence of Zn$^{2+}$. **F** 2D [$^1$H,$^{15}$N] HSQC spectrum of $^{15}$N-labeled I3$_{1-78}$SSS alone (black) and upon addition of a 4-fold molar excess of ZnSO$_4$ (cyan). **G** ITC thermograms of I3$_{1-78}$ with ZnSO$_4$. Data are presented as mean values ± SD; $n = 4$ technical replicates. **H** ITC thermograms of full-length I3$_{1-126}$ with ZnSO$_4$. Data are presented as mean values ± SD; $n = 4$ technical replicates.

identical to wildtype (K$_D$, 13.1 ± 2.8 nM to 16.8 ± 2.8 nM; Table 1 and S1; Fig. S2). This demonstrates that the I3 non-canonical SILK motif, $^{28}$LTIK$^{31}$, insignificantly contributes to PP1 binding when PP1 is already anchored by the RVxF and the CCC motifs, likely due to the lack of a canonical Ile residue at the 'I' position (SILK; the Ile residues normally binds deeply into the SILK hydrophobic pocket). However, as our NMR data shows, the I3 SILK motif contributes to PP1 binding when the I3 RVxF motif is deleted (Fig. 1E, F). We then quantified the contribution of the CCC motif by deleting I3 residues 60–68 (I3$_{27-59}$; CCC$_{deletion}$; Fig. 3A). The affinity of the I3 CCC$_{deletion}$ variant for PP1 was ~40-fold weaker than WT (K$_D$, 376 ± 47 nM, Table 1 and S1; Fig. 3B). This change in affinity was accompanied by a change in the binding kinetics profiles. Indeed, I3$_{27-59}$ binding to PP1 results in an SPR sensorgram profile typical of most IDP PP1 interactors, such as the PP1 regulator NIPP1; namely, it is fit with a single exponential with fast on- and off-rates[14]. These data show that the biphasic binding kinetics of I3$_{27-68}$ with PP1 is likely due to the interaction of the CCC motif at PP1 active site. Accordingly, I3$_{27-59}$ also inhibits PP1 poorly (IC$_{50}$ = 422 ± 113 nM, Table 1 and S1, Fig. 3C). These data demonstrate the importance of the CCC motif for both PP1 binding and inhibition. Finally, to determine if the

residue identity of the CCC motif is essential and/or if the residues C-terminal to Cys62$_{I3}$ are important, we tested three additional variants for their ability to bind and inhibit PP1: I3$_{1-126}$SSS ($^{60}$CCC$^{62}$→$^{60}$SSS$^{62}$), I3$_{27-68}$SSS ($^{60}$CCC$^{62}$→$^{60}$SSS$^{62}$) and I3$_{27-68}$ IY ($^{63}$IY$^{64}$→$^{63}$AA$^{64}$). While I3$_{27-68}$ IY behaves like WT I3$_{27-68}$ (K$_D$, 15.6 ± 2.4 nM, Table 1 and S1; Fig. S2), the I3$_{27-68}$ SSS variant is statistically identical to the CCC deletion variant, I3$_{27-59}$ (K$_D$, 412 ± 6 nM, IC$_{50}$ = 380 ± 31 nM, Table 1 and S1, Fig. 3C, D). Together, these data demonstrate that the I3 $^{60}$CCC$^{62}$ motif is essential for both PP1 binding and inhibition.

**Dissecting the roles of the CCC motif cysteines**

We next tested the importance of each Cys residue in the I3 CCC motif by generating all combinations of C→S single and double variants (I3$_{27-68}$: SCC, CSC, CCS, CSS, SCS, SCC; Fig. 3A). We then tested their ability to bind and inhibit PP1. Each single C→S variant bound PP1 with similar affinities (K$_D$ 18.8–21.7 nM; Fig. 3E; Table 1 and S1) that were ~2.0-fold weaker than wt (K$_D$ 9.3 nM). They also exhibited weaker inhibition (2.1–3.2-fold lower inhibition; Fig. 3C; Table 1 and S1). While a similar affinity was observed for the CSS double variant (K$_D$ of 21.2 nM),

double variants with $Cys60_{I3}$ mutated to serine (SSC and SCS) bound PP1 8.9 and 5.6-fold more weakly ($K_D$s of 86.6 and 54.0 nM, respectively; Fig. 3F; Table 1 and S1). The inhibitory capacity of these variants was also weaker. The $IC_{50}$ for the CSS and SCS variants were similar to those of the single point variants (2.1 and 2.4-fold higher $IC_{50}$; Fig. 3G; Table 1 and S1). In contrast, the $IC_{50}$ value for the SSC variant was statistically identical to those of the $CCC_{deletion}$ and SSS variants (4.6-fold higher $IC_{50}$; Fig. 3G; Table 1 and S1). Taken together, these data highlight that it is the number of cysteines present in the CCC motif, and not their position within the motif, that is important for PP1 binding and inhibition, as the loss of a single cysteine at any position is largely tolerated. More significant impacts on PP1 binding and especially inhibition were observed only when either two or all three cysteines were mutated. Furthermore, these data showed that $Cys60_{I3}$ is especially important, as the double mutant retaining this cysteine (CSS) had $K_D$ and $IC_{50}$ values similar to those of the single mutant variants. In contrast, the double variants with the Cys60Ser mutation (SCS, SSC) had $K_D$ and $IC_{50}$ values that were significantly higher. Notably, while the SSC variants had an $IC_{50}$ value statistically identical to that of the SSS and $CCC_{deletion}$ variants, its binding affinity was still stronger, demonstrating that the presence of a single cysteine enhances binding by nearly 5-fold compared to the SSS variant. Together, these data demonstrate that the CCC motif constitutes a dynamic, fuzzy binding interaction of I3 at the PP1 active site that is necessary for PP1 binding and inhibition.

## I3 binding at the PP1 active site

Our data show that the identity of the residue in the CCC motif, namely a sulfur containing cysteine, is critical for PP1 binding and inhibition. One unique function of cysteines is their ability to form disulfide bonds. Although no electron density consistent with a disulfide bond between any of the CCC motif cysteines and PP1 was observed, $Cys273_{PP1}$, which is the second most reactive cysteine in PP1[43] and forms a covalent S-C bond with the small molecule active site inhibitor microcystin-LR[2], is close to the active site and the CCC motif cysteines. Further, both Cys273 and the CCC motif have some of the highest B-factors in the I3:PP1 complex (Fig. 4A). To determine if Cys273 is important for I3 binding and inhibition, we generated the PP1 C273S variant and tested its interaction with $I3_{27-68}$. The affinity of $I3_{27-68}$ for PP1 C273S ($K_D$, $16.1 \pm 0.1$ nM) was nearly identical to wt PP1 ($9.3 \pm 1.1$ nM), as was its inhibition ($IC_{50}$: PP1 C273S, $85 \pm 3.0$ nM; WT, $71.8 \pm 2.7$; assays performed in the presence of 1.33 mM DTT). Consistent with these data, the $IC_{50}$ of $I3_{27-68}$ for PP1 was statistically identical in the absence of DTT ($IC_{50}$, $67.5 \pm 6.3$, no DTT; Fig. S5). Together, these data confirm that $Cys273_{PP1}$ disulfide bond formation is not important for PP1 binding and inhibition by I3 (Fig. 4B, C, Table 1 and S1).

PP1 has two metals bound at the active site that are strictly required for catalysis (Fig. 2E). If one or both metals are missing, PP1 is inactive. Mammalian PP1 typically contains a $Zn^{2+}$ (M1) and a $Fe^{2+}$ (M2) ion at the active site[44]. Bacterial PP1 purified from *E. coli*, which is supplemented with $MnCl_2$ during expression, contains two $Mn^{2+}$ ions at the active site[2,45]. M1 is coordinated by three PP1 residues, His66, Asp64 and Asp92, while M2 is coordinated by four PP1 residues, Asp92, Asn124, His173 and His248[2]. M1 is labile and readily exchanges from the active site in solution, while M2 is less likely to exchange, reflecting its additional side chain interaction with PP1[14]. In addition to forming disulfide bonds, cysteines are also important for coordinating metals[46]. To test if the I3 CCC motif interacts with metal ions, we used NMR spectroscopy and isothermal titration calorimetry (ITC). First, we recorded a 2D [$^1$H,$^{15}$N] HSQC spectrum of $^{15}$N-labeled $I3_{1-78}$ and then added $ZnSO_4$. I3 residues $^{50}$EHMGRRSSKCCCI$^{63}$, which include the CCC motif, exhibited reduced intensities, demonstrating that these residues interact directly with $Zn^{2+}$ (Fig. 4D, E). When the same experiment was performed using a variant of I3 in which the CCC motif cysteines

were mutated to serines (CCC to SSS, $I3^{1-78}$SSS), the intensities of these residues were unaffected by the presence of $Zn^{2+}$ (Fig. 4F). These data show that the CCC motif cysteines are necessary and sufficient for binding $Zn^{2+}$. To determine the binding affinity of I3 for $Zn^{2+}$, we used ITC, which showed that both $I3_{1-126}$ and $I3_{1-78}$ bind $Zn^{2+}$ with identical affinities ($I3_{1-126}$ $K_D = 3.2 \pm 0.7$ μM; $I3_{1-78}$ $K_D = 3.0 \pm 0.4$ μM; Fig. 4G, H; Table S3).

After establishing that I3 binds metal, we then tested if the binding of I3 to PP1 is sensitive dependent on the presence of both metals, M1 and M2, in PP1 active site. We generated three PP1 variants shown or predicted to disrupt the binding of either M1 or M2 to PP1[14,47]. First, we tested the PP1 substrate trapping variant H66K. This variant is both folded and stable, yet is unable to bind the M1 metal (the H66K lysine sidechain prevents M1 metal binding)[14]. SPR affinity measurements showed that $I3_{27-68}$ binds WT and PP1 H66K with identical affinities ($K_D$, $5.9 \pm 0.6$ nM, Table 1 and S1; Fig. S2), demonstrating that the loss of the M1 metal does not impact I3 binding. We then generated constructs predicted to inhibit M2 binding (H248N and H248K)[47]. H248N failed to inhibit M2 metal binding and thus bound I3 with wt affinity ($K_D$, $6.6 \pm$ nM; Table 1 and S1, Fig. S2). H248K failed to purify, demonstrating that a metal at the M2 position is required for PP1 stability. We also examined the PP1 active site in the I3:PP1 structure. In both the reconstituted and co-expressed I3:PP1 complexes, both metals are present at the catalytic site. The I3:PP1 electron density shows that the I3 CCC motif clearly binds in front of the active site; however, the density is weaker than the rest of I3 and more similar to that observed for the I3 $^{54}$GRRSS$^{58}$ motif and the PP1 Cys273 loop. Taken together, the NMR, crystallography and SPR data show that the CCC motif binds in a dynamic, fuzzy manner at the PP1 active site, instead of a single conformation, which is essential for both PP1 binding and inhibition.

## Discussion

Multiple members of the PPP family of phosphatases, including PP1, are regulated by small, IDP protein-based inhibitors that potently inhibit the activity of their cognate PPP. These inhibitors of PP1 are Inhibitor 1 (I1, DARPP-32)[48,49], Inhibitor 2 (I2)[50] and Inhibitor 3 (I3)[7]. PP1 is also potently inhibited by a folded regulator, SDS22[14,15]. Despite their common ability to inhibit PP1, the emerging view is that the mechanisms by which these evolutionary ancient inhibitors achieve PP1 inhibition differ. For example, while I1/DARPP-32 potently inhibits PP1 when it is phosphorylated by protein kinase A (PKA; phosphorylates Thr34; the DARRP-32:PP1 complex structure has not yet been determined and thus will not be discussed further)[51], SDS22, I2 and I3 achieve PP1 inhibition in the absence of phosphorylation. Understanding the distinct mechanisms used by these inhibitors to bind and inhibit PP1 is vital for gaining insights into the distinct processes that these inhibitors mediate in PP1 biology.

Previously, we showed that SDS22 inhibits PP1 by blocking putative substrate recognition grooves (the hydrophobic and C-terminal substrate binding grooves) and by stabilizing a conformation of PP1 that lacks its M1 metal[14]. Because this metal is essential for the activation of the nucleophilic water that mediates phosphate hydrolysis, PP1 bound to SDS22 is completely inactive. In this way, SDS22 facilitates PP1 cellular activity by serving as a PP1 cellular 'storage' protein, holding PP1 in an inactive conformation until it is needed for holoenzyme assembly. I2 inhibits PP1 differently[52]. In addition to binding PP1 via a SILK and RVxF motif, I2 also binds PP1 via an extended inhibitory α-helix (I2 residues 130–169; this helix is prominently prepopulated in free I2)[33]. The I2 inhibitory helix binds across the PP1 active site via multiple interactions at the PP1 acidic and hydrophobic grooves (Fig. S6A). This binding event positions the side chains of $His148_{I2}$, $Tyr149_{I2}$ and $Glu151_{I2}$ to bind and block the PP1 active site (Fig. S6B). Thus, while I2 is an IDP in its unbound form, it binds PP1 in a single conformation and, via its inhibitory helix, blocks access to the PP1 active site thereby potently inhibiting PP1 activity.

Here, we show that while I3 shares several PP1 interaction mechanisms with I2 (i.e., they both bind PP1 via SILK and RVxF motifs and also bind the PP1 acidic groove, Fig. S6C), the mechanism by which I3 inhibits PP1 differs yet again. Namely, I3 uses three sequential cysteine residues (CCC motif) to, in a dynamic/'fuzzy' manner, bind and block the PP1 active site, likely via transient engagement with the active site metals. Multiple lines of evidence support this finding. First, the identity of the residues in this motif (Cys) is critical for PP1 binding and inhibition, as their conservative mutation to serine is equivalent to deleting them entirely (see binding and inhibition data for $CCC_{deletion}$ vs SSS, Table 1). Second, our SPR and activity data show that the loss of any of the three cysteines has minimal, yet identical impacts on PP1 binding and inhibition, demonstrating that the remaining cysteines, independent of their position in the I3 polypeptide chain, readily compensates for single C → S mutation. Third, this binding and inhibition is not mediated by an intermolecular disulfide bond, as I3 binds and inhibits both PP1 and $PP1_{C273S}$ with equivalent affinities and $IC_{50}$ values, respectively. Fourth, ITC and NMR data show that I3, via the CCC motif, binds metals. Because a PP1 variant that lacks the M1 metal, $PP1_{H66K}$, binds I3 with an affinity equivalent to PP1 (Table 1), our data also show that only a single PP1 active site metal (M2) is required to achieve the observed I3 binding and inhibition. Although it is established that dynamic, fuzzy interactions that are most commonly resulting from charge:charge interactions are critical mediators of protein:protein binding, providing selectivity and affinity beyond single conformation protein:protein interactions[26], our data show that dynamic/fuzzy interactions can also be leveraged for enzyme inhibition. To the best of our knowledge this is the initial example of such a dynamic/fuzzy interaction that results in enzyme inhibition.

In addition to their distinct mechanisms of PP1 inhibition, I2 and I3 are also unusual in their ability to form distinct PP1 triple complexes. I2 readily forms a triple complex with spinophilin (spinophilin:I2:PP1)[23], another PP1 regulator that, like I2, also forms a dimeric complex with PP1 (spinophilin:PP1)[27]. Similarly, I3 forms a triple complex with SDS22 (SDS22:I3:PP1)[11,53]. However, how these complexes form differ. Overlaying the I3:PP1 and SDS22:PP1 structures (Fig. S6D) using PP1 shows that SDS22 and I3 bind PP1 via distinct interfaces, allowing both proteins to bind simultaneously to PP1 without altering the conformations observed in their dimeric complexes (Fig. S6E). In contrast, overlaying the I2:PP1 and spinophilin:PP1 complexes shows that I2 and spinophilin bind PP1 via overlapping surfaces: both I2 and spinophilin bind PP1 using an RVxF motif, one of which must release in the spinophilin:I2:PP1 triple complex. NMR studies showed that, in the triple complex, it is the spinophilin RVxF motif that stays bound to PP1 in the spinophilin:I2:PP1 triple complex[23]. That is, the non-canonical I2 RVxF motif (SQKW) is readily displaced that of spinophilin because it has a polar 'Q' residue in the position normally occupied by a hydrophobic 'V'. I2 is able to remain associated due to the additional interactions made via its SILK and inhibitory α-helix.

I3, like I2, also has a non-canonical RVxF motif, KVEW. While the major motif residues (K, V, W) are canonical, the 'E' is not. This is because more than 50% of all RVxF-containing PP1 regulators have an 'S' or 'T' at the 'x' position and phosphorylation of these residues (an event mimicked by mutagenesis to a 'D' or 'E') greatly attenuates PP1 binding in vitro and in vivo[20,21]. Thus, I3, like I2, has a weak RVxF motif compared to other PP1 IDP regulators. However, like I2, our data show that I3 is also able to bind PP1 in the absence of a functional RVxF motif. What role this weak I3 RVxF interaction may or may not play in PP1 holoenzyme assembly from SDS22:I3:PP1 complexes is an active area of investigation. Finally, while the assembly of many PP1 holoenzymes is controlled, in part, by phosphorylation status of the 'x' residue in regulator RVxF motifs, the assembly of PP1 inhibitory complexes is clearly not: SDS22 does not contain an RVxF motif,

while the RVxF motifs of DARPP32, I2 and I3 do not contain phosphorylatable residues at the 'x' position. Thus, global increases in cellular phosphorylation inhibit PP1 activity via two mechanisms: it blocks the assembly of scores of PP1 holoenzymes due to the phosphorylation of regulator RVxF motifs while, simultaneously, PP1:inhibitor complexes are either not sensitive (SDS22, I2, I3) or enhanced (DARPP-32) by phosphorylation. In conclusion, our data not only show how the distinct phosphorylation susceptibilities of PP1 targeting versus inhibitory proteins facilitate PP1 inhibition in cells, but critically reveals a different mechanism of PP1 inhibition, in which a dynamic, fuzzy interaction of the I3 CCC motif at the PP1 active site potently blocks PP1 activity.

## Methods

### Bacterial protein expression

The coding sequence of human Inhibitor-3 (I3) containing an N-terminal TEV (tobacco etch virus) protease cleavage site was synthesized by DNA 2.0. The coding sequences of human $I3_{1-126}$, $I3_{27-126}$ and $I3_{38-126}$ were subcloned into a pET-M30-MBP vector containing an N-terminal his6-tag followed by maltose binding protein (MBP) and a TEV (tobacco etch virus) protease cleavage site. *Escherichia coli* BL21 (DE3) cells (Agilent) were transformed with I3 expression vectors. Freshly transformed cells were grown at 37 °C in LB broth containing kanamycin antibiotics (50 μg/ml) until they reached an optical density ($OD_{600}$) of 0.6–0.8. Protein expression was induced by addition of 1 mM β-D-thiogalactopyranoside (IPTG) to the culture medium, and cultures were allowed to grow overnight (18–20 h) at 18 °C. Cells were harvested by centrifugation (6000 × $g$, 15 min, 4 °C) and stored at −80 °C until purification. Expression of uniformly $^{13}C$- and/or $^{15}N$-labeled protein was carried out by growing freshly transformed cells in M9 minimal media containing 4 g/L [$^{13}C$]-D-glucose and/or 1 g/L $^{15}NH_4Cl$ (Cambridge Isotopes Laboratories) as the sole carbon and nitrogen sources, respectively. I3 variants $I3_{1-78}$, $I3_{27-68}$, $I3_{27-59}$ and $I3_{38-68}$ were generated by introduction of a stop codon after the respective site; $I3_{1-126}$ $^{40}KAEA^{43}$, $I3_{1-126}$ C60S/C61S/C62S, $I3_{1-126}$ T29A/I30A/K31A, $I3_{1-78}$ C60S/C61S/C62S, $I3_{27-68}$ C60S, $I3_{27-68}$ C61S, $I3_{27-68}$ C62S, $I3_{27-68}$ C60S/C61S, $I3_{27-68}$ C60S/C62S, $I3_{27-68}$ C61S/C62S, $I3_{27-68}$ C60S/C61S/C62S, $I3_{27-68}$ I63A/Y64A and $I3_{27-68}$ T29A/I30A/K31A were generated by site-directed mutagenesis, sequence verified and expressed as described above. Cloning and expression of $PP1α_{1-330}$, $PP1α_{7-300}$, $PP1α_{7-330}$, $PP1α_{7-330}$ C273S, $PP1α_{7-330}$ H66K, $PP1α_{7-330}$ H248N and $PP1α_{7-330}$ H248K was performed as previously described[30,35]. The $I3_{79-91}$ peptide was synthesized (Bio-Synthesis Inc).

### Mammalian protein expression

$PP1α_{7-300}$ and GFP-$I3_{27-68}$ were cloned into mammalian expression vector pcDNA3.4_K_RP1B with an N-terminal His6-tag and a TEV cleavage site. The plasmids were amplified and purified using the NucleoBond Xtra Maxi Plus EF (Macherey-Nagel). $PP1α_{7-300}$ and GFP-$I3_{27-68}$ were co-expressed in Expi293F cells (ThermoFisher; cat# A14527) at a 1:1 ratio of 1.0 μg DNA per mL of final transfection culture volume. Transfections were performed using 500 mL medium (Expi293 Expression Medium, ThermoFisher) in 2 L flasks according to the manufacturer's protocol in an incubator at 37 °C and 8.5% $CO_2$ under shaking (125 rpm). On the day of transfection, the cell density was adjusted to $2×10^6$ cells/mL using fresh Expi293 expression medium. DNA of PP1 and I3 (1:1 ratio) were diluted in Opti-MEM Reduced Serum Medium (ThermoFisher). Similarly, in a separate tube, 3× the amount of PEI (Polysciences) was diluted in the same volume of Opti-MEM Reduced Serum Medium (ThermoFisher). The DNA and PEI mixtures were combined and incubated for 10 min, before added into the cell culture. Valproic acid (2.2 mM final concentration, Sigma) was added to the cells 4 h after transfection. The cells were harvested 72 h by centrifugation (2000 × $g$, 20 min, 4 °C) and the pellet was stored at −80 °C.

## Protein purification

Cell pellets expressing I3 (I3$_{1-126}$, I3$_{1-78}$, I3$_{27-68}$, I3$_{27-59}$, I3$_{38-68}$, I3$_{1-126}$ $^{40}$KAEA$^{43}$, I3$_{1-126}$ C60S/C61S/C62S, I3$_{1-126}$ T29A/I30A/K31A, I3$_{1-78}$ C60S/C61S/C62S, I3$_{27-68}$ C60S, I3$_{27-68}$ C61S, I3$_{27-68}$ C62S, I3$_{27-68}$ C60S/C61S, I3$_{27-68}$ C60S/C62S, I3$_{27-68}$ C61S/C62S, I3$_{27-68}$ C60S/C61S/C62S, I3$_{27-68}$ I63A/Y64A and I3$_{27-68}$ T29A/I30A/K31A) were resuspended in ice-cold lysis buffer (50 mM Tris pH 8.0, 500 mM NaCl, 5 mM imidazole, 0.1% Triton X-100 and an EDTA-free protease inhibitor tablet [Roche]) and lysed by high pressure homogenization (Avestin EmulsiFlex C3). Lysate was clarified by centrifugation (45,000 × $g$, 45 min, 4 °C) and the supernatant was loaded under gravity onto Ni$^{2+}$-NTA beads (GE Healthcare) pre-equilibrated with 50 mM Tris pH 8.0, 500 mM NaCl and 5 mM imidazole. Protein was eluted in 4 column volumes (20 ml) using 50 mM Tris 8.0, 500 mM NaCl and 500 mM imidazole. Fractions with I3 were pooled and dialyzed with TEV protease overnight at 4 °C against 50 mM Tris pH 8.0, 500 mM NaCl and 20 mM DTT to cleave the MBP-His$_6$-tag. The cleaved protein was heat purified at 95 °C (15 min), filtered using a 0.22 μm filter (Millipore), concentrated and further purified using size-exclusion chromatography (SEC; Superdex 75 26/60 [Cytiva]) equilibrated in either NMR Buffer (20 mM Bis-Tris pH 6.8, 500 mM NaCl, 0.5 mM TCEP), SEC/SPR buffer (20 mM Tris pH 8.0, 500 mM NaCl, 0.5 mM TCEP and 1 mM MnCl$_2$) or ITC buffer (20 mM Tris pH 8.0, 500 mM NaCl, 0.5 mM TCEP). SEC fractions were pooled, concentrated and stored at −20 °C.

PP1 was purified as described previously[30]. Briefly, PP1 (PP1α$_{1-300}$, PP1α$_{7-300}$, PP1α$_{7-330}$, PP1α$_{7-330}$ C273S, PP1α$_{7-330}$ H66K, PP1α$_{7-330}$ H248N and PP1α$_{7-330}$ H248K) was lysed in PP1 Lysis Buffer (25 mM Tris pH 8.0, 700 mM NaCl, 5 mM imidazole, 1 mM MnCl$_2$, 0.1% Triton X-100), clarified by centrifugation (45,000 × $g$, 45 min, 4 °C) and immobilized on Ni$^{2+}$-NTA resin. Bound His$_6$-PP1 was washed with PP1 Buffer A (25 mM Tris pH 8.0, 700 mM NaCl, 5 mM imidazole, 1 mM MnCl$_2$), followed with a stringent wash containing 6% PP1 Buffer B (25 mM Tris pH 8.0, 700 mM NaCl, 250 mM imidazole, 1 mM MnCl$_2$) at 4 °C. The protein was eluted using PP1 Buffer B and His$_6$-tagged PP1 was purified using SEC (Superdex 200 26/60 [Cytiva]) pre-equilibrated in SEC Buffer (20 mM Tris pH 8, 500 mM NaCl, 0.5 mM TCEP, 1 mM MnCl$_2$). Peak fractions were incubated overnight with TEV protease at 4 °C. The cleaved protein was incubated with Ni$^{2+}$-NTA beads (GE Healthcare), the flow-through collected and immediately used. All experiments were performed with freshly purified PP1. All PP1 variants were purified to homogeneity except PP1α$_{7-330}$ H248K, which was unstable and immediately precipitated.

## I3$_{1-126}$:PP1α$_{1-330}$, I3$_{1-126}$ $^{40}$KAEA$^{43}$:PP1α$_{1-330}$ (for NMR) and I3$_{27-68}$:PP1α$_{7-300}$ complex formation (for crystallography)

For NMR spectroscopy the I3$_{1-126}$:PP1α$_{1-330}$ complex was established as follows: purified PP1α$_{1-330}$ was mixed with an excess of $^{15}$N-labeled I3$_{1-126}$, concentrated and the complex purified using SEC (NMR buffer). The I3$_{1-126}$ $^{40}$KAEA$^{43}$:PP1α$_{1-330}$ complex for NMR spectroscopy was constituted by adding $^{15}$N-labeled I3$_{1-126}$ $^{40}$KAEA$^{43}$ (1 mM stock solution) to purified PP1α$_{1-330}$ in a 1:1 ratio. The I3$_{1-126}$ $^{40}$KAEA$^{43}$:PP1α$_{1-330}$ complex was concentrated (0.1 mM) and directly used for NMR experiments. Crystallography of the reconstituted I3$_{27-68}$:PP1α$_{7-300}$ complex was established as follows: PP1 was incubated with 1.5-fold excess of I3$_{27-68}$ and the complex was purified using SEC (crystallization buffer). The final reconstituted I3$_{27-68}$:PP1α$_{7-300}$ complex concentration was ~7 mg/mL that was used directly for crystallization trials (vapor diffusion/sitting drop).

## NMR spectroscopy

NMR experiments were acquired on a Bruker Neo 600 MHz spectrometer, equipped with a TCI HCN-z cryoprobe at 283 K. NMR spectra for the assignment of I3$_{1-126}$ and I3$_{1-78}$ were acquired using $^{15}$N,$^{13}$C-labeled protein at a final concentration of 0.25 mM in 20 mM Bis-Tris pH 5.6, 100 mM NaCl, 0.5 mM TCEP and 90% H$_2$O/10% D$_2$O. The following spectra were used to complete the sequence specific backbone assignments: I3$_{1-126}$, 2D [$^1$H,$^{15}$N] HSQC, 3D HNCA, 3D HN(CO)CA, 3D HNCACB, 3D CBCA(CO)NH, 3D (HA)CANCO and 3D HNCO; I3$_{1-78}$, 2D [$^1$H,$^{15}$N] HSQC, 3D HNCA, 3D HN(CO)CA, 3D HNCACB and 3D CBCA(CO)NH. All spectra were processed using Topspin 4.1.1 (Bruker, Billerica, MA), and Cara (http://cara.nmr.ch) was used for the sequence-specific chemical shift assignments.

For I3$_{1-126}$ all residues except Arg35, Arg55, Lys59, Cys61, Lys66, Glu80 and Glu81 have been assigned. The interaction between I3$_{1-126}$ and I3$_{1-126}$ KAEA with PP1α$_{1-330}$ was studied by direct comparison of the 2D [$^1$H,$^{15}$N] HSQC spectra of free $^{15}$N-labeled I3$_{1-126}$ and I3$_{1-126}$ $^{40}$KAEA$^{43}$, and in complex with PP1α$_{1-330}$. The final concentration of the complex was 0.1 mM in NMR buffer and 90% H$_2$O/10% D$_2$O. The interaction between I3$_{1-78}$ and I3$_{1-78}$ SSS with Zn$^{2+}$ was studied by direct comparison of the 2D [$^1$H,$^{15}$N] HSQC spectra of $^{15}$N-labeled I3$_{1-78}$ or $^{15}$N-labeled I3$_{1-78}$ C60S/C61S/C62S with and without a 4-fold molar excess of ZnSO$_4$. The final concentration of the complex was 0.1 mM in NMR buffer at pH 5.8 with 90% H$_2$O/10% D$_2$O. The spectra were processed using Topspin 4.1 and analyzed using NMRFAM-Sparky[54].

## Crystallization and structure determination of the I3$_{27-68}$:PP1α$_{7-300}$ complex

The mammalian co-expressed I3$_{27-68}$:PP1α$_{7-300}$ complex (6.5 mg/ml) crystalized in condition 1–2 of the Morpheus Screen, (Molecular Dimensions; 0.1 M imidazole; 0.1 M MES acid pH 6.5, 20% (v/v) ethylene glycol; 10% PEG8000 and divalents [0.06 M Magnesium chloride hexahydrate; 0.06 M Calcium chloride dihydrate]). The bacterially expressed and reconstituted I3$_{27-68}$:PP1α$_{7-300}$ complex (10 mg/ml) crystallized in condition 2–10 of the Morpheus Screen (Molecular Dimensions; 0.1 M TRIS-base; 0.1 M bicine pH 8.5, 20% (v/v) ethylene glycol; 10% PEG8000 and ethylene glycols [0.12 M Diethylene glycol; 0.12 M Triethylene glycol; 0.12 M Tetraethylene glycol; 0.12 M Pentaethylene glycol]). Crystals were cryo-protected in silicone oil and then flash frozen in liquid N$_2$. Diffraction data were collected at Beamline SSRL 12-2 and processed using XDS[55] and autoxds[56]. The structures of the I3$_{27-68}$:PP1α$_{7-300}$ complexes were determined by molecular replacement using Phaser as implemented in PHENIX[57]. PP1 (4MOV) was used as the search model[30]. A solution was obtained in space group P4$_1$4$_1$2 for both complexes. The model was completed using iterative rounds of refinement in PHENIX and manual building using Coot[58].

## Surface plasmon resonance (SPR)

SPR measurements were performed using a 4-channel Reichert 4SPR instrument fitted with autosampler and a degassing pump (Reichert Technologies). SPR buffers containing 20 mM Tris pH 8.0, 500 mM NaCl, 1 mM MnCl$_2$, 0.5 mM TCEP, 0.05% Tween-20 were prepared, sterile filtered, and degassed in autoclaved glassware prior to each experiment. Running buffer was used to prime and run both the sample and syringe pump reservoirs. Gold sensorchips modified with Ni-NTA-functionalized polycarboxylate (NiHC200M; XanTec Bioanalytics GmbH) were installed and equilibrated under flow conditions (100 μL/min) for ≥60 min at 25 °C. Surface contaminants were cleared from the chip surface by a pair of 120 μL injections of 2 M NaCl and 10 mM NaOH during the equilibration step. Experiments were conducted at 25 °C with a 5 Hz sampling rate and were initiated by injecting 180 μL of His$_6$-PP1 (PP1α$_{7-330}$, PP1α$_{7-330}$ C273S, PP1α$_{7-330}$ H66K and PP1α$_{7-330}$ H248N) constructs (40–80 nM) diluted in 20 mM Tris pH 8.0, 500 mM NaCl, 1 mM MnCl$_2$, 0.5 mM TCEP, 0.05% Tween-20 onto channels 1, 2 and 3 for 180 s at 50 μL/min which resulted in between 200 and 450 μRIU of surface loading (channel 4 was used as reference). The sensorchip was allowed to equilibrate for 5 min at 50 μL/min prior to initiation of experiments. The concentrations of I3$_{1-126}$ and its variants (I3$_{1-126}$ $^{40}$KAEA$^{43}$, I3$_{1-126}$ C60S/C61S/C62S, I3$_{1-126}$ T29A/I30A/K31A, I3$_{27-68}$, I3$_{38-68}$, I3$_{27-59}$, I3$_{27-68}$ I63A/Y64A, I3$_{27-68}$ C60S, I3$_{27-68}$ C61S, I3$_{27-68}$ C62S, I3$_{27-68}$ C60S/C61S, I3$_{27-68}$ C60S/C62S, I3$_{27-68}$ C61S/C62S and

I3$_{27-68}$ C60S/C61S/C62S, I3$_{27-68}$ T29A/I30A/K31A, I3$_{79-91}$) were measured using AccuOrange™ Protein Quantification Kit (Biotium). For measurements, I3$_{1-126}$ and its variants were diluted into running buffer from concentrated stocks, and a series of injections at different I3 concentrations were applied. A total of 60–120 μL sample of I3 were respectively injected for 60–120 s at 50 μL/min followed by a dissociation step of 120–300 s. For all experiments, buffer blank injections were included at an interval of two sample injections to achieve double referencing. Technical replicates were obtained by utilizing three channels per chip coupled with stripping of the sensorchip with 350 mM EDTA pH 8.0, reconditioning the surface with 10 mM NaOH to remove non-specifically bound PP1 aggregates, charging the surface with 10 mM NiSO$_4$, and reloading fresh PP1 onto the surface. All replicates were generated with freshly diluted PP1 and I3. Kinetic parameters were determined by curve-fitting using TraceDrawer software (Ridgeview Instruments AB) fit with a one-to-one one state or one-to-one two state model. Statistical analyses of SPR data were performed using Microsoft Excel.

### Isothermal Titration Calorimetry (ITC)

ZnSO$_4$ (100 μM; 10 μL per injection) was titrated into purified I3$_{1-126}$ or I3$_{1-78}$ (6–10 μM) in ITC buffer (20 mM Tris pH 8.0, 500 mM NaCl, 0.5 mM TCEP) at 25 °C using VP-ITC (Malvern). Each experiment consisted of 25 injections, with 250 sec intervals between injections to allow for complete equilibration and baseline recovery; the solution in the sample cell was stirred at 300 rpm to ensure rapid mixing. Thermodynamic (ΔH, ΔS, ΔG) and binding constant (K$_a$) data were determined using NITPIC[59], SEDPHAT[60] and GUSSI[60].

### IC$_{50}$ assays

The IC$_{50}$ values of I3 variants with PP1α were determined using the phosphorylated peptide H3pT3 (Histone 3 pT3: NH2-AR[pT] KQTARKSTGGKAPRKQLA-COOH, BioSynthesis) as substrate and the PiColorLock Gold reagent (Novus Biologicals) for the detection of the released Pi. In each 96 well (Costar), 5 μL of serially diluted I3 was added to 75 μL of PP1 (1.2 nM PP1 in 40 mM HEPES pH 7.0, 0.013% (v/v) Tween-20, 0.133 mg/mL BSA, 1.33 mM sodium ascorbate; 1.33 mM DTT was used for all assays except that testing the effect of DTT on inhibition, where DTT was omitted from the assay) and pre-incubated for 30 min at room temperature. The high-signal controls were prepared by adding buffer instead of I3 to PP1. The low signal controls were the buffer only without PP1. The reaction was initiated by the addition of 20 μL of 250 μM substrate H3pT3 using a multichannel pipette and incubated at 30 °C for 30 min. The reactions were terminated by adding 25 μL PiColorLock reagent with its accelerator. A total of 10 μL of stabilizer was added before the absorbance was measured at 635 nm using CLARIOstar (BMG Labtech). The percentage of PP1 activity was calculated using the following equation: [(absorbance-low signal control)/(high-signal control−low-signal control)] × 100%. IC$_{50}$ curves were fitted using the 4-parameter logistic equation (Prism 9.0).

### Reporting summary

Further information on research design is available in the Nature Portfolio Reporting Summary linked to this article.

## Data availability

The NMR data generated in this study have been deposited in the BioMagResBank database under accession code BMRB 51565 and 51566. The atomic coordinates and structure factors generated in this study have been deposited in the PDB database under accession code 8DWK and 8DWL. All ITC/SPR and IC$_{50}$ data generated in this study are provided in the Supplementary Information and/or Source Data file, which is available at Figshare [https://doi.org/10.6084/m9.figshare.22133528]. Source data are provided with this paper.

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

## Acknowledgements

We thank Drs. Kelly Tatchell and Lucy Robinson (Louisiana State University Health Sciences Center) for fruitful discussions and critical reading of the manuscript. We acknowledge the initial efforts for crystallization of the I3:PP1 holoenzyme by Drs. Sarah Sheftic and Ganesan Senthil Kumar. We would like to thank staff members at the Stanford Synchrotron Radiation Light source (SSRL), SLAC National Accelerator Laboratory for access to X-ray beamlines. Use of the Stanford Synchrotron Radiation Lightsource, SLAC National Accelerator Laboratory, is supported by the U.S. Department of Energy, Office of Science, Office of Basic Energy Sciences under Contract No. DE-AC02-76SF00515. This work was supported by grant 1R01GM144483 from the National Institute of General Medicine and R01NS124666 from the National Institute of Neurological Disorders and Stroke to W.P. and grant R01GM098482 from the National Institute of General Medicine to R.P.

## Author contributions

R.P. and W.P. developed the concept. G.S. and M.S.C. expressed and purified all proteins. G.S., M.S.C., and N.C.B. performed NMR

experiments. M.S.C. and G.S. crystallized and determined the structures of I3:PP1 complexes. G.S. performed all SPR experiments. N.C.B. performed ITC experiments. R.P. and W.P. wrote the manuscript with comments and inputs from all co-authors.

## Competing interests

The authors declare no competing interests. The funders had no role in study design, data collection and analysis, decision to publish, or preparation of the manuscript.
