## [Peer Review File · Nature Communications]

REVIEWER COMMENTS

Reviewer #1 (Remarks to the Author):

The manuscript of Srivastava and coworkers reports a study on the structure of a complex formed between protein phosphatase-1 catalytic subunit (PP1) and inhibitor-3 (I3). I3 is expressed in all lineages of eukaryotes and it is an ancient inhibitor of PP1 with putative roles in PP1 maturation and regulation. The structure of many PP1-regulator/inhibitor complex (PP1-I2, PP1-spinophilin, PP1-MYPT1 etc.) have been solved, but the detailed features of PP1-I3 interaction has not been revealed so far. In this manuscript the authors have provided compelling description of the PP1-I3 interactions using wide spectra of „up to date” methods (NMR, kinetic assays, SPR, ITC and X-ray crystallography) for interaction analyses. First, they analyzed the PP1-I3 interactions with NMR and surface plasmon resonance (SPR) based binding technique and dissected the major regions of I3 (residues 27-68) responsible for binding of PP1. This truncated I3 showed identical binding and inhibitory effectiveness to the wild type I3 (1-126). The analyses of the PP1-I3 (27-68) crystals identified interaction of PP1 with RVxF and SILK binding motifs, as well as with two basic patches and a cysteine reach regions (61CCC63). One of the basic patches occupies the substrate binding acidic groove in PP1 and it also serves to direct cysteines toward the catalytic sites for inhibition. The latter inhibitory action is a novel way of exertion of inhibition by I3 in a “fuzzy” manner, not found with other PP1 inhibitors. The authors hypothesize that this cysteine reach region may interact with the M2 metal ion (Fe²⁺ or Mn²⁺) in PP1; however, this claim is not fully supported by the presented data. Overall, this study provides significant novel data on the regulation PP1 by I3 and further extend our knowledge on the structure-activity relationships of the interaction of the PP1 family with inhibitory and regulatory proteins.

Although, the experimentation and the presentation of the manuscript appears to be quite compact a few critical notes may be considered.

Major concerns:

1. The authors should make it straightforward if the SILK-like motif of I3 has significant role in binding to PP1c. The binding experiments show that the “SILK-like motif dead” mutant has the same affinity to PP1c as the wild-type I3. These data question any role of this region influencing I3 binding to PP1 and this should be discussed in a more objective manner.
2. There is a “dual like presentation” of the RVxF PP1 binding motif of I3 in the manuscript. First, the authors describe the KVEW sequence as a canonical RVxF motif, however, later on they claim that this sequence is not “so canonical” due to the presence of E in the sequence which could work against interaction with PP1, therefore, they consider this sequence as a weak interaction site. Assuming the experimental results of the binding studies the KVEW sequence appears to play an important role in the interaction, so the discussion its role as a weaker RVxF binding motif should be described in a more precise way.
3. The authors hypothesize that the “CCC” sequence in I3 acts through interaction with the M2 metal ion in PP1. They claim that the interaction of WT I3 and its truncated form interacts with Zn²⁺ and this makes it likely that this region of I3 may interact with M2 metal ion. This conclusion, however, seems to be a little farfetched. First, M2 metal ion in PP1 could be Fe²⁺ in mammalian cells and Mn²⁺ in PP1 obtained from bacterial expression. Therefore, proving interaction of I3 with Zn²⁺ (a candidate for M1 metal ion) is not plausible to prove interaction of I3 “CCC” sequence with M2 in PP1c even in an indirect way. In addition, it is not obvious from the ITC experiment that how many Zn²⁺ (n?) binds to I3.

Minor notes:

1. Page 5, row before last: “extended confirmation”?
2. Page 6, in the subtitle at the end of page: SLIK or SILK?
3. Page 15, roll?, supposed to be role?

Reviewer #2 (Remarks to the Author):

Protein phosphatase 1 (PP1) is a ubiquitous and highly significant enzyme that catalyzes a vast set of dephosphorylation events under the control of ~200 co-regulatory proteins. Among the most ancient and well conserved inhibitors of PP1 is Inhibitor-3, which regulates PP1 through a mechanism that, surprisingly, remains largely undescribed at the molecular scale. Here, the authors combine structural, thermodynamic, and enzymatic analysis to reveal the role of three key regions within Inhibitor-3 toward regulation of PP1. The novelty of the interactions observed, relative to other characterized PP1 inhibitors, and the broad thematic similarity to the investigator's own observations from a history of studying other protein phosphatases (i.e., multiple short linear interactions across both regulatory sites and the active site) combine to make an engaging and highly important study.

The experiments performed are reported rigorously, with adequate attention to statistical analysis, and with complete deposition of relevant data into public databases. As a result, the comments here largely reflect questions of interpretation or suggestions for improving the readability of the manuscript (especially for the non-expert in protein phosphatases).

1. When reading the introduction, in conjunction with Figure 1, it was very confusing to see the amino acid sequence KVEW labeled as an RVxF motif. While K-R and W-F are both conservative mutations, this sequence difference nevertheless left me wondering how well conserved the so-called RVxF motif actually is. The discussion does a nice job of describing the relative level of conservation in this motif across related proteins and clarifies this point. Would it be helpful to briefly comment on the expectation of sequence conservation in the introduction as well? As an aside that supports this point, the first discussion of the SILK motif is accompanied by a very clear discussion of conservation within this motif.

2. The final sentence of the introduction teases that the observed inhibition mechanism "could be shared within all phosphoprotein phosphatase family members." The meaning of this statement is ambiguous. Are the authors referring to the necessity for not one but two (or more) SLiM interactions with a single inhibitor (i.e., a common mechanism with RCAN1/CN) or the interaction mediated by the Cys-Cys-Cys motif? If it is the latter, how prevalent are these motifs across protein phosphatase co-regulators and are they well conserved across species when they are encountered?

3. On page 11, where the results from the PP1 C273S mutant are discussed, equilibrium dissociation constant data are provided, but no inhibition study or IC50 are reported, which is unusual in the context of the data presented. Am I simply missing the relevant inhibition study? If not, Then it is not clear the authors can claim that "disulfide bond formation is not important for PPI binding and inhibition by I3" (emphasis added).

4. It is not clear from the data presented why the Cys residues are required (although the requirement for them is beyond question, based on the study presented). The most obvious explanation that is not ruled out is presented in the discussion: that one or more makes a direct (or at least water-mediated) interaction with the metals in the active site, most likely M1. Yet, the crystal structure does not seem to support this, at least not from the views shown in the figures. Certainly, Cys residues would be poised to interact with a Zn²⁺ in the M1 site, as is suggested by the Inhibitor-3 metal binding studies. Is PPI capable of binding other metals to site 1 that would be expected to have different affinity for the CCC sequence? This question could quickly become speculative and demand an out-of-scope inquiry, but it dwells on the least satisfying aspect of the current work.

5. Minor typographical errors:

The superscript '2' is missing in the second author affiliation.

The introduction paragraph starting with "Here use molecular and cell biology..." is poorly written, relative to the rest of the manuscript, containing a string of at least four typographical errors in the first six lines.

On page 6, the second to last sentence appears to be missing a reference to Figure 1H.

Reviewer #3 (Remarks to the Author):

In the present manuscript Wolfgang Peti et al describe how inhibitor 3 interacts with PP1 through different sequence motifs and inhibit PP1 activity. Like Inhibitor 2, Inhibitor 3 is binding to PP1 via a SILK and RVxF motifs, but it does inhibit PP1 with a different mechanism. The author suggests that three consecutive cysteines from inhibitor 3 bind PP1 active site in a dynamic way and engage with the active site metals, preventing substrate access and dephosphorylation. They identified a new type of interaction between a CCC motif and PP1 active site and conduct a thorough structural and biophysical analysis. They carry out SPR, NMR and IC50 studies to investigate the binding in more detail. The work has been well carried out and the data support the author conclusion. Their findings help to further understand how PP1 is regulated, and they described here a new mechanism of PP1 inhibition with a motif which has not been identified previously.

I think that the results presented here are of a great interest in the field.

I have however one major concern that would need to be addressed and some minor comments, mainly about the figures.

Major comment which needs clarification:

Page 11: The author discussed the potential role of Cys273PP1 in I3 binding:
"To determine if Cys273 is important for I3 binding and inhibition, we generated PP1 C273S and tested its interaction with I327-68. The affinity of I327-68 for PP1 C273S (KD, 15.1 ± 4 nM) was nearly identical to wt PP1 (9.3 ± 1.1 nM), confirming that C273PP1 disulfide bond formation is not important for PP1 binding and inhibition by I3 (Fig. 4B. Table 1)."

By looking at the material and method section, PP1 is purified in the presence of 0.5 mM TCEP and the buffer used for the affinity measurement by SPR also contains 0.5 mM TCEP which is highly likely to prevent the formation of any disulphide bridge. If the authors want to test whether C273 is forming a disulphide bridge with a cysteine of the CCC motif, then they should repeat the affinity measurement of the wildtype I3 in absence of TCEP and with protein purified without TCEP. They also used 1.33mM DTT in their IC50 assay, so they might want to do an IC50 measurement for I31-126 wt or I327-68 wt in the absence of DTT and with proteins not purified in the presence of TCEP and compare with their current data.

Minor concerns/comments:

Title: I'm not sure about the use of the word "fuzzy" in the title, dynamic might be more appropriate.

Page 4: Here "we" used molecular and cell biology... -> Word "we" missing

Page 5: The authors are discussing Kon and Koff within PP1 regulators and the fact that I3 shows biphasic kinetics with a fast and slow dissociation event. Would it be possible

to extract both K_D s and both k_{on}/k_{off} from their SPR data?

Page 13: Remove "the" before "acidic groove" in "Here, we show that while I3 shares several PP1 interaction mechanisms with I2 (i.e., they both bind PP1 via SILK and RVxF motifs and also bind the PP1 the acidic groove, Fig. S5C)..."

Figures:

It is relatively hard to follow the structure analysis/description in the results part by looking at the figures provided in the manuscript, particularly figure 2. The authors should make sure that the figure color coding is consistent throughout the manuscript and show + label all residues discussed in the result part.

Figure 2 is not very clear and need to be improved.

- It would improve clarity to move the density map in a supplementary figure.
- The yellow I3 on top of beige/white surfaces and the yellow labelling is making it very difficult to see details. A better contrast would improve clarity.
- There are also some inconsistencies in the labels colour: blue vs yellow for I3 (ex: DTVDNE in 2A and 2C).
- Some residues discussed in the text aren't labelled (M53, K260 and A259 in 2C).
- 2D shows electrostatic surface of PP1 but 2C and the overview below it (bottom right) seems to show a pocket manually coloured in red in pymol for PP1 surface underneath I355-59 (pocket colored in beige in 2A-left) and white for the rest of PP1. If the authors want to discuss PP1-I3 electrostatic interaction in 2C and 2D why they don't use the same colour code (full electrostatic surface) for 2C and 2D and the bottom right should be the same or white or as 2A...

Figure 3A: The last double mutant should be labelled CSS and not SSC

Figure 4F and 4G: the K_D s units are missing (μM).

Figure S2: Add Asn86 which is discussed in the text.

Figure S3: PP1 is displayed in white surface in all figures except S3. It would make sense to change it to white.

Figure S4: K_D of PP17-330:I63A/Y64A I3 27-68 (Bottom left) is 15.5 on the figure and 15.6 in the text (page 9).

Figure S5B: Labelling of H148 missing

Figure S5D: Mn atoms displayed in red instead of purple in all other figures and it might be a good idea to use a different color for SDS22 as green is the color of caspase in figure S3

Reviewer #4 (Remarks to the Author):

The manuscript by Srivastava and colleagues describes the structural and biophysical aspects of Protein Phosphatase 1 inhibition by Inhibitor-3. The mechanistic aspects of this interaction differ from other PP1 inhibitors, but more importantly show how different combinations of interaction motifs can combine to achieve biological function. The crux of the paper is the description of a novel CCC motif that inhibits the active site and forms a fuzzy interaction with metal ions. The presented research is interesting and merits a publication, if CCC motif indeed makes fuzzy interaction with metal ions. However In my opinion some aspects of the paper, particularly the fuzzy binding of CCC motifs should be supported by additional data, given that such a mode of interaction has not been observed before.

Main issues:

I have some reservations regarding the conclusion that CCC makes fuzzy interactions

with metal ions, although it seems that several experiments indeed point in this direction. However, since the authors suggest that this is one of the main findings of the paper, the case should really be solid. Here are a few comments regarding the CCC motif:

-main point: after establishing that I3 and I3(27-68) have identical affinity and activity most of the experiments are made on the I3(27-68) construct. In particular, the mutations C>S dissecting the role of CCC motif are also made in truncated version of I3 (I327-68). In this construct the CCC motif is near the C-end of polypeptide, but in the full-length I3 the motif is in the center of the chain. I think it is necessary to confirm some of the mutagenesis results on the full-length I3. This is because higher dynamics/fuzziness of CCC motifs could also be the consequence of the motif being placed at the chain end, since it is known that chain termini tend to have increased flexibility. The data on figure 1e show that residues beyond 68 have reduced intensity, thus they may interact with the PP1 and make the CCC motif less flexible. The bottom line is that some of the mutants in Figure 3 need to be verified in the context of full-length I3, to ensure that fuzziness is not a consequence of increased dynamics due to placing the CCC motif to the C-terminus.

-second point: NMR experiments were performed at low pH (pH<7), SPR binding experiments at high pH (pH>7). (For ITC I cannot find in which buffer they were made, see comment below). The pKa of cysteine is in a range of 7-8, which suggests that cysteine may have had different protonation state in these experiments. Is there a particular reason for using high pH in binding experiments? How does the protonation state affect interactions of CCC motif? It is well known that reactivity of cysteine dependent on pH, due to linked equilibrium. I suggest that authors verify how pH affects the CCC motif interaction as this might importantly enhance the understanding of nature of this interaction.

-are there any NOEs for the residues near the CCC motif and PP1?

-PRE-NMR would be very useful to get a more detailed picture of the CCC-PP1 interactions, if metal ion is substituted to a paramagnetic ion (as in recombinant protein) this could be used to obtain a direct measurement of CCC-active site distances.

-Is CCC motif conserved in I3 homologs? Perhaps this is missing in the discussion, if the fuzzy interactions with CCC motifs and metal are something more general or a unique feature of I3

-there is very limited information provided on the structure of CCC motif based on the crystallographic data. The structure of the shorter I3-PP1 complex is described in the second paragraph of the results section. Interactions between all motifs are described in detail, but the structural details for the interactions between CCC motif and PP1 are practically omitted. What are the distances between cysteines and metal ions? Is the problem because the B-factors are very high in this region? Is it possible to conclude anything from the crystal structure regarding the interactions of CCC with PP1, if not why? On figure 4C, for example how far are thiol groups from metal? Can interaction be expected based on the distance?

-page 6. Reduction of intensities was used to locate the regions of I3 that bind to PP1. Authors conclude that 2 regions in RVxF dead construct interact with PP1. However the data on figure 1E also shows a third region (around residue 80) with reduced intensities. Can you comment on that and explain why this region does not interact with PP1?

-design of 'dead' mutations. There is one point in the design and interpretation of mutant effects that is a bit misleading. This is the issue with deletions or mutations of motifs. I don't think that deletion of motifs (eg. CCC dead) and SSS mutant are directly comparable. This is because for IDPs it is often the case that chain entropy plays an important role in resulting affinity. To state simply interactions by amino acids make favourable interactions, but backbone gives unfavourable entropic contribution. When you mutate residues you remove interactions but retain backbone/entropic component. However in deletions you remove both, that is why deletions can have much less pronounced effects than mutations of motifs. This is relevant for your conclusion regarding the importance of SILK vs. RVxF motifs. The SILKdead variant has identical affinity compared to wt, which is likely because chain deletion was used in this case. On the other hand for RVxF motifs its functionality was inspected via KAEA mutation. If the functionality of these two motifs is compared that an appropriate mutation of SILK is

needed.

-ITC: what is the buffer used for ITC experiments?

-ITC: it would make sense to make a control titration with SSS variant of some of the S/C variants to verify that cysteine indeed bind metals. The NMR data (Figure 4E) shows a reduction of intensities also for the residues before CCC motif (50-60). Could be Histidine 53 also be involved in binding? Histidine and cysteine are well known to coordinate metal ions.

-ITC: please note that enthalpies in Table S3 should be considered as apparent, since cysteine-metal ion interaction likely involves exchange of protons, which cause significant enthalpy effect. This is why experiments are usually performed in buffer with low enthalpy of ionization.

Minor :

-Table 1: In my opinion Table 1 legend should contain also the following information: temperature (to which Kd values refer) and explanation of Fit and n columns.

-Table 1: number of digits. Errors and the corresponding parameters are given with either 1, 2 or 3 significant digits. All errors should be rounded to either 1 or 2 significant digits uniformly and the corresponding parameters should be rounded accordingly.

-Figure 1H: I found the description (FL) unnecessary and confusing in figure legend, since FL acronym is not really used in the text

-Figure 1A caption. What is the meaning of the underlined sequence is not explained.

-Figure 4 caption: 'Sensogram between ... and ... '. Sentence is incomplete?.

-Figure 4 panel C : there is some text 'molar ratio' that does not belong here

-Figure 4 panels F and G missing units for the Kd. Panel G missing x-axis label

We would like to thank all four reviewers for their careful review of the manuscript and very much appreciate their support for our work. It was exciting to see that all four reviewers agreed that our work is well-performed and of high interest. As you will see, we have revised the manuscript to answer all questions of the reviewers, including a large number of additional experiments, which clearly improved the manuscript. Thus, we feel that we have fully addressed all the points raised. We hope that you will find the article with the included changes suitable for publication in *Nature Communications*.

In the response to the comments/questions, we have done the following to make our responses easy to read and identify:

- included all the Reviewer comments in **bold italics**.
- Our responses are listed immediately below each point in standard font.

Reviewer #1 (Remarks to the Author):

The manuscript of Srivastava and coworkers reports a study on the structure of a complex formed between protein phosphatase-1 catalytic subunit (PP1) and inhibitor-3 (I3). I3 is expressed in all lineages of eukaryotes and it is an ancient inhibitor of PP1 with putative roles in PP1 maturation and regulation. The structure of many PP1-regulator/inhibitor complex (PP1-I2, PP1-spinophilin, PP1-MYPT1 etc.) have been solved, but the detailed features of PP1-I3 interaction has not been revealed so far. In this manuscript the authors have provided compelling description of the PP1-I3 interactions using wide spectra of „up to date” methods (NMR, kinetic assays, SPR, ITC and X-ray crystallography) for interaction analyses. First, they analyzed the PP1-I3 interactions with NMR and surface plasmon resonance (SPR) based binding technique and dissected the major regions of I3 (residues 27-68) responsible for binding of PP1. This truncated I3 showed identical binding and inhibitory effectiveness to the wild type I3 (1-126). The analyses of the PP1-I3 (27-68) crystals identified interaction of PP1 with RVxF and SILK binding motifs, as well as with two basic patches and a cysteine reach regions (61CCC63). One of the basic patches occupies the substrate binding acidic groove in PP1 and it also serves to direct cysteines toward the catalytic sites for inhibition. The latter inhibitory action is a novel way of exertion of inhibition by I3 in a “fuzzy” manner, not found with other PP1 inhibitors. The

authors hypothesize that this cysteine reach region may interact with the M2 metal ion (Fe²⁺ or Mn²⁺) in PP1; however, this claim is not fully supported by the presented data. Overall, this study provides significant novel data on the regulation PP1 by I3 and further extend our knowledge on the structure-activity relationships of the interaction of the PP1 family with inhibitory and regulatory proteins.

We thank the reviewer for her/his strong support, including our “compelling description of the PP1-I3 interactions”, the “significant novel data on the regulation PP1 by I3”, the “novel way of exertion of inhibition” and “further extend our knowledge on the structure-activity relationships of the interaction of the PP1 family with inhibitory and regulatory proteins”.

Although, the experimentation and the presentation of the manuscript appears to be quite compact a few critical notes may be considered.

Major concerns:

1. The authors should make it straightforward if the SILK-like motif of I3 has significant role in binding to PP1c. The binding experiments show that the “SILK-like motif dead” mutant has the same affinity to PP1c as the wild-type I3. These data question any role of this region influencing I3 binding to PP1 and this should be discussed in a more objective manner.

We agree with the reviewer that the I3 SILK motif does not contribute significantly to the overall binding affinity of I3 to PP1. However, this does not mean it does not play a role in cellular function. Rather, it may be similar to the SILK motif of I2, which adopts a more prominent role in the triple complex formed between spinophilin:I2:PP1 (1). Like I2, I3 also forms a triple complex, the SDS22:I3:PP1 triple complex, where it may also facilitate the recognition of I3 by AAA ATPases, as reported by the laboratory of Dr. Hemmo Meyer (see (2)). Confirmation of these potential functions is outside the scope of the current manuscript, but our data of the I3 RVxF dead variant (see Figure 1F in current manuscript) when bound to PP1 shows strong binding of the SILK site. This suggests a similar function for those motifs as that in I2.

Additionally, combined with the input of reviewer 4, we have now also performed further experiments. We generated the SLTIK→SLAAA (instead of the deletion construct) in both I3₂₇₋₆₈ and full-length I3. We then repeated the SPR experiments and tested for PP1 binding. The experiments showed similar results as the SILK_{deletion} mutant regarding binding strength (K_D) of I3 with PP1. We have now included these new data into the revised manuscript.

2. There is a “dual like presentation” of the RVxF PP1 binding motif of I3 in the manuscript. First, the authors describe the KVEW sequence as a canonical RVxF motif, however, later on they claim that this sequence is not “so canonical” due to the presence of E in the sequence which could work against interaction with PP1, therefore, they consider this sequence as a weak interaction site. Assuming the experimental results of the binding studies the KVEW sequence appears to play an important role in the interaction, so the discussion its role as a weaker RVxF binding motif should be described in a more precise way.

We thank the reviewer for highlighting this inconsistency – we updated the manuscript to reflect that while the I3 RVxF motif fits the standard RVxF definition, it is still atypical as it cannot be modified by phosphorylation on the x residue, which usually leads to a significant weakening of the interaction with PP1.

Like most PP1 regulators, I3 is predicted to be an IDP, is heat stable and binds PP1 using a PP1 RVxF SLiM (⁴⁰KVEW⁴³; **Fig. 1A**) (18). The hydrophobic residues V and F are the key RVxF binding elements, with R/K and F/W identified as common substitutions in biochemically confirmed RVxF motifs (19). Typically, the 'x' residue in the RVxF motif is an S or T and, when phosphorylated, significantly weakens PP1 binding (20). This weakening is also achieved by mutating the 'x' residue to a phosphomimetic D/E (21). Thus, the 'E' at the 'x' position of the RVxF motif is atypical and, furthermore, makes the phosphorylation-induced disruption of RVxF binding impossible for I3.

3. The authors hypothesize that the “CCC” sequence in I3 acts through interaction with the M2 metal ion in PP1. They claim that the interaction of WT I3 and its truncated form interacts with Zn²⁺ and this makes it likely that this region of I3 may interact with M2 metal ion. This conclusion, however, seems to be a little farfetched. First, M2 metal ion in PP1 could be Fe²⁺ in mammalian cells and Mn²⁺ in PP1 obtained from bacterial expression. Therefore, proving interaction of I3 with Zn²⁺ (a candidate for M1 metal ion) is not plausible to prove interaction of I3 “CCC” sequence with M2 in PP1c even in an indirect way. In addition, it is not obvious from the ITC experiment that how many Zn²⁺ (n?) binds to I3.

We agree with the reviewer that, based on our data, the scientific conclusion is that the CCC motif interacts with the M2 metal ion at the PP1 active site. We used multiple distinct, complimentary techniques to experimentally support the conclusion that this interaction is dynamic (fuzzy) and cannot be captured using a single method, i.e., X-ray crystallography.

The NMR spectroscopy data demonstrating that Zn²⁺ binds I3 was used to demonstrate that bivalent metal ions are coordinated by the I3 CCC motif. This experiment was not used to show the interaction of the CCC with the M2 metal. It has been reported in the literature that the M2 metal in PP1 is typically either Mn²⁺ (for bacterially expressed protein) or Fe²⁺ (for HEK293F expressed protein). Recently, we experimentally determined PP1 complex structures that contain two Zn²⁺ ions at the active site when expressed in HEK293F cells suggesting that under certain cellular conditions, Zn²⁺ can substitute for Fe²⁺ at the M2 site. Consistent with this result, PP1 purified from rabbit muscle also contains 74% Zn²⁺ and only 8% Fe²⁺ (3, 4). Both Fe²⁺ and Mn²⁺ are paramagnetic, i.e., have an unpaired electron. Zn²⁺ is diamagnetic. Thus, NMR measurements with Zn²⁺ can be readily used to directly monitor divalent metal binding, while such measurements performed with paramagnetic Fe²⁺ or Mn²⁺ are very difficult to interpret as changes in the I3 spectrum can be due to a direct metal ion binding event or just due to through space paramagnetic relaxation enhancement.

To summarize, these specific NMR measurements were not meant to test the metal type in the M2 site; rather, the use of Zn²⁺ was necessary to allow for clear NMR data interpretation if

I3 can interact with a divalent metal ion. That is, we used Zn^{2+} because it allows for rigorous data interpretation.

Minor notes:

1. Page 5, row before last: “extended confirmation”?

Thank you – updated.

2. Page 6, in the subtitle at the end of page: SLIK or SILK?

Thank you – updated.

3. Page 15, roll?, supposed to be role?

Thank you – updated.

Reviewer #2 (Remarks to the Author):

Protein phosphatase 1 (PP1) is a ubiquitous and highly significant enzyme that catalyzes a vast set of dephosphorylation events under the control of ~200 co-regulatory proteins. Among the most ancient and well conserved inhibitors of PP1 is Inhibitor-3, which regulates PP1 through a mechanism that, surprisingly, remains largely undescribed at the molecular scale. Here, the authors combine structural, thermodynamic, and enzymatic analysis to reveal the role of three key regions within Inhibitor-3 toward regulation of PP1. The novelty of the interactions observed, relative to other characterized PP1 inhibitors, and the broad thematic similarity to the investigator’s own observations from a history of studying other protein phosphatases (i.e., multiple short linear interactions across both regulatory sites and the active site) combine to make an engaging and highly important study.

We very much appreciate the strong support of reviewer 2.

The experiments performed are reported rigorously, with adequate attention to statistical analysis, and with complete deposition of relevant data into public databases. As a result, the comments here largely reflect questions of interpretation or suggestions for improving the readability of the manuscript (especially for the non-expert in protein phosphatases).

1. When reading the introduction, in conjunction with Figure 1, it was very confusing to see the amino acid sequence KVEW labeled as an RVxF motif. While K-R and W-F are both conservative mutations, this sequence difference nevertheless left me wondering how well conserved the so-called RVxF motif actually is. The discussion does a nice job of describing the relative level of conservation in this motif across related proteins and clarifies this point. Would it be helpful to briefly comment on the expectation of sequence conservation in the introduction as well? As an aside that supports this point, the first discussion of the SILK motif is accompanied by a very clear discussion of conservation within this motif.

We appreciate that reviewer 2 highlights this issue – which was clearly written in a manner assuming the audience are PP1 experts. We have updated the manuscript accordingly to highlight that the RVxF motif of I3 falls within the accepted definition of RVxF motifs (thus well conserved). We have added information from the discussion when writing about the RVxF motif in the result section to clarify the definition for the readers.

We have added this text to the introduction (see also the response to reviewer 1):

Like most PP1 regulators, I3 is predicted to be an IDP, is heat stable and binds PP1 using a PP1 RVxF SLiM (⁴⁰KVEW⁴³; **Fig. 1A**) (18). The hydrophobic residues V and F are the key RVxF binding elements, with R/K and F/W identified as common substitutions in biochemically confirmed RVxF motifs (19). Typically, the 'x' residue in the RVxF motif is an S or T and, when phosphorylated, significantly weakens PP1 binding (20). This weakening is also achieved by mutating the 'x' residue to a phosphomimetic D/E (21). Thus, the 'E' at the 'x' position of the RVxF motif is atypical and, furthermore, makes the phosphorylation-induced disruption of RVxF binding impossible for I3.

2. The final sentence of the introduction teases that the observed inhibition mechanism “could be shared within all phosphoprotein phosphatase family members.” The meaning of this statement is ambiguous. Are the authors referring to the necessity for not one but two (or more) SLiM interactions with a single inhibitor (i.e., a common mechanism with RCAN1/CN) or the interaction mediated by the Cys-Cys-Cys motif? If it is the latter, how prevalent are these motifs across protein phosphatase co-regulators and are they well conserved across species when they are encountered?

We thank the reviewer for highlighting the lack of accuracy in our presentation. We intended to highlight that *the mode of inhibition using the CCC motif* could be shared by other PPP family members and not the use of multiple SLiM interaction with a single inhibitor. As highlighted by the reviewer, the use of multiple SLiM interactions within a single inhibitor is well-established for PP1 and CN (PP3).

We have updated the statement for improved clarity:

Together, this study provides novel insights into the mechanism by which I3 inhibits PP1, furthers our understanding of the physiological regulation of PP1 and highlights a likely novel inhibitory interaction via a CCC motif that might be shared between all phosphoprotein phosphatase family members.

A search for other proteins with a CCC motif using ProSite shows that the CCC motif is found 3438 times in 2594 of 568744 UniProtKB/Swiss-Prot sequences (0.46% of sequences searched). Thus, while the motif is rare, it is clearly present in other proteins. Currently, it is impossible to predict if they function in a manner similar to I3. Thus, our statement was meant to highlight the possibility for future investigations.

3. On page 11, where the results from the PP1 C273S mutant are discussed, equilibrium dissociation constant data are provided, but no inhibition study or IC₅₀ are reported, which is unusual in the context of the data presented. Am I simply missing the relevant inhibition study? If not, Then it is not clear the authors can claim that “disulfide bond formation is not important for PPI binding and inhibition by I3” (emphasis added).

Thank you for highlighting this omission. We have performed the requested IC₅₀ experiment using PP1_{C273S}, which shows that the IC₅₀ is unchanged, confirming our conclusion that disulfide bond formation is not important for PP1 binding and inhibition. Consistent with this result, the measured IC₅₀ of I3₂₇₋₆₈ with PP1 in the presence and absence of DTT is also identical. These data have been added to the revised manuscript in Table 1, Table S1, Figure 4C and Figure S5

4. It is not clear from the data presented why the Cys residues are required (although the requirement for them is beyond question, based on the study presented). The most obvious explanation that is not ruled out is presented in the discussion: that one or more makes a direct (or at least water-mediated) interaction with the metals in the active site, most likely M1. Yet, the crystal structure does not seem to support this, at least not from the views shown in the figures. Certainly, Cys residues would be poised to interact with a Zn²⁺ in the M1 site, as is suggested by the Inhibitor-3 metal binding studies. Is PPI capable of binding other metals to site 1 that would be expected to have different affinity for the CCC sequence? This question could quickly become speculative and demand an out-of-scope inquiry, but it dwells on the least satisfying aspect of the current work.

We agree with the reviewer that the most obvious explanation for the function of the cysteine residues is that one or multiple cysteine residues make a direct or water-mediated interactions with the metal ions at the PP1 active site. However, we have ruled out that this interaction is with the M1 metal. Namely, to test for a possible M1 interaction, we used the PP1 variant H66K. As we previously showed (5), PP1_{H66K} does not bind an M1 metal as the lysine residue intrudes into the M1 binding space (steric), as well as it lacks the ability to coordinate the metal ion in a manner that a His residue can (chemistry); these statements are based on the PP1_{H66K} crystal structure and multiple assays. Thus, PP1_{H66K} is a PP1 catalytically inactive variant of PP1 that lacks the M1 metal (indeed, it has started to become a widely used tool for many in the PP1 field). However, we show that I3 still binds PP1_{H66K} with the same affinity as with wt-PP1. These data show that the M1 metal is not required for I3 CCC motif binding with PP1.

Unfortunately, despite us and others trying for many years, a stable PP1 variant that allows the exclusion of the M2 metal in the PP1 active site has not yet been identified (mutations of M2 metal coordinating residues fail to fold). Several variants were proposed by Dr. Ernie Y Lee (multiple manuscripts from 1995-1998; and tested again for this study). M2 is coordinated by 4 residues while M1 is coordinated by 3. To generate an M2 deficient PP1, we focused on mutations of PP1 residue H248, as it functions in a similar manner as H66 for M1. However, the amino acids

substituted were still either able to bind a metal (H248N; while not of high quality due to the low stability of this PP1 variant, SPR data reports an identical K_D with I3) or their yield and stability were exceedingly low (H248K) making SPR measurements impossible to perform. This highlights the importance of the M2 metal not only for the catalytic activity of PP1, but moreover for PP1 overall stability. We agree with the reviewer that this is the least satisfying aspect of the work – but we have tried for many, many years to find a stable PP1 variant without M2 – at this point we do not think it exists, as binding of a metal ion to M2 is strictly required for correct folding of PP1.

5. Minor typographical errors:

The superscript ‘2’ is missing in the second author affiliation.

Thank you – updated.

The introduction paragraph starting with “Here use molecular and cell biology...” is poorly written, relative to the rest of the manuscript, containing a string of at least four typographical errors in the first six lines.

Thank you – updated.

On page 6, the second to last sentence appears to be missing a reference to Figure 1H.

Thank you – updated.

Reviewer #3 (Remarks to the Author):

In the present manuscript Wolfgang Peti et al describe how inhibitor 3 interacts with PP1 through different sequence motifs and inhibit PP1 activity. Like Inhibitor 2, Inhibitor 3 is binding to PP1 via a SILK and RVxF motifs, but it does inhibit PP1 with a different mechanism. The author suggests that three consecutive cysteines from inhibitor 3 bind PP1 active site in a dynamic way and engage with the active site metals, preventing substrate access and dephosphorylation. They identified a new type of interaction between a CCC motif and PP1 active site and conduct a thorough structural and biophysical analysis. They carry out SPR, NMR and IC50 studies to investigate the binding in more detail. The work has been well carried out and the data support the author conclusion. Their findings help to further understand how PP1 is regulated, and they described here a new mechanism of PP1 inhibition with a motif which has not been identified previously.

I think that the results presented here are of a great interest in the field.

We appreciate the strong support of reviewer 3.

I have however one major concern that would need to be addressed and some minor comments, mainly about the figures.

Major comment which needs clarification:

Page 11: The author discussed the potential role of Cys273PP1 in I3 binding:

“To determine if Cys273 is important for I3 binding and inhibition, we generated PP1 C273S and tested its interaction with I327-68. The affinity of I327-68 for PP1 C273S (KD, 15.1 ± 4 nM) was nearly identical to wt PP1 (9.3 ± 1.1 nM), confirming that C273PP1 disulfide bond formation is not important for PP1 binding and inhibition by I3 (Fig. 4B. Table 1).”

By looking at the material and method section, PP1 is purified in the presence of 0.5 mM TCEP and the buffer used for the affinity measurement by SPR also contains 0.5 mM TCEP which is highly likely to prevent the formation of any disulphide bridge. If the authors want to test whether C273 is forming a disulphide bridge with a cysteine of the CCC motif, then they should repeat the affinity measurement of the wildtype I3 in absence of TCEP and with protein purified without TCEP. They also used 1.33mM DTT in their IC50 assay, so they might want to do an IC50 measurement for I31-126 wt or I327-68 wt in the absence of DTT and with proteins not purified in the presence of TCEP and compare with their current data.

As requested by the reviewer, we have performed the experiments, which we agree are a necessary control. They did not show a change and thus disulfide formation is not contributing to binding or inhibition. Furthermore, addition or removal of DTT, as suggested by the reviewer, did not change the inhibition. The data is summarized in Figure R1 and is now included in the manuscript as Figure S5.

Fig R1: Effect of DTT and the role of PP1_{C273S} for inhibition of I3. A. IC₅₀ measurement of I3₂₇₋₆₈ in the presence or absence of 1.33 mM DTT. **B.** IC₅₀ measurement of I3₂₇₋₆₈ with WT and PP1_{C273S}. IC₅₀ measurements were performed in buffer containing 1.33 mM DTT. Error bar = standard deviation, n = 8-12.

Minor concerns/comments:

Title: I'm not sure about the use of the word “fuzzy” in the title, dynamic might be more appropriate.

We agree with the reviewer – although ‘fuzzy’ is now accepted lexicon for these types of interactions, we are also not ‘fans’ of the word “fuzzy”, as it not scientifically precise. Dynamic is certainly more appropriate from a scientific point of view. We have changed “fuzzy” to dynamic – thank you.

Page 4: Here “we” used molecular and cell biology... -> Word “we” missing

Thank you – updated.

Page 5: The authors are discussing K_{on} and K_{off} within PP1 regulators and the fact that I3 shows biphasic kinetics with a fast and slow dissociation event. Would it be possible to extract both K_D s and both k_{on}/k_{off} from their SPR data?

We performed a rigorous global data fit using all available data; we also recorded our data using a 4-channel SPR instrument (Reichert) to allow for continuous referencing. However, even under such rigorous conditions, fitting individual K_D s from these data would lead to untrustworthy data interpretation.

Thus, while we do not distinguish the K_D s, we do understand the differences of the interaction – as shown when either I3 27-59 or I3 27-68 SSS is used for the SPR measurements, we see a single k_{on}/k_{off} event, highlighting that SILK/RVxF etc. binding is fast single step on/off event, as was seen with other PP1 regulators. Thus, the slow off rate is most likely due to the dynamic/fuzzy CCC motif interaction.

Page 13: Remove “the” before “acidic groove” in “Here, we show that while I3 shares several PP1 interaction mechanisms with I2 (i.e., they both bind PP1 via SILK and RVxF motifs and also bind the PP1 the acidic groove, Fig. S5C)...”

Thank you – updated.

Figures:

It is relatively hard to follow the structure analysis/description in the results part by looking at the figures provided in the manuscript, particularly figure 2. The authors should make sure that the figure color coding is consistent throughout the manuscript and show + label all residues discussed in the result part.

We thank the reviewer for the careful input regarding the Figures, which have all been implemented and the figures have been carefully updated.

Figure 2 is not very clear and need to be improved.

- It would improve clarity to move the density map in a supplementary figure.
- The yellow I3 on top of beige/white surfaces and the yellow labelling is making it very difficult to see details. A better contrast would improve clarity.
- There are also some inconsistencies in the labels colour: blue vs yellow for I3 (ex: DTVDNE in 2A and 2C).
- Some residues discussed in the text aren't labelled (M53, K260 and A259 in 2C).
- 2D shows electrostatic surface of PP1 but 2C and the overview below it (bottom right) seems to show a pocket manually coloured in red in pymol for PP1 surface underneath I355-59 (pocket colored in beige in 2A-left) and white for the rest of PP1. If the authors want to discuss PP1-I3 electrostatic interaction in 2C and 2D why they don't use the same colour

code (full electrostatic surface) for 2C and 2D and the bottom right should be the same or white or as 2A...

In response to the reviewer's suggestions, we have completely overhauled Figure 2. It now incorporates all suggestions and additional changes to improve clarity.

- The electron density figure has been removed (since the structure and map can be downloaded from the RCSB) and replaced with other figures of the complex that we believe illustrate the interaction more clearly.
- Figure 1I includes an 'overall' structure figure with I3 shown as yellow sticks and the distinct binding pockets on PP1 discussed throughout the text colored in pink to green shades. The % solvent accessible surface area per I3 residue is shown in Figure 1J and the same pockets highlighted.
- Figure 2A shows I3 alone to more clearly show the I3 conformation.
- Figure 2B shows the electrostatic surface of I3 with the N-terminal portion of I3. The PP1 residues that define the acidic pocket, along with polar interactions with I3 are shown in Figure 2C.
- Figure 2D shows the I3 residues that mediate the 'kink' towards the active site with the caspase DTVD residues shown in a slightly darker shade of yellow to distinguish them from the other I3 residues (but within the same pallet for I3). The polar/electrostatic interactions with PP1 are indicated.
- Figure 2E shows, again, the electrostatic surface potential of PP1 with I3 shown as sticks (showing the deep acidic binding groove of PP1), but this time viewing the C-terminal portion of I3. The PP1 and I3 residues that define the acidic groove and their interactions with I3 residues are shown in Figure 2F.

Figure 3A: The last double mutant should be labelled CSS and not SSC.

This has been corrected.

Figure 4F and 4G: the kDs units are missing (uM).

This has been corrected.

Figure S2: Add Asn86 which is discussed in the text.

N86 has been added to Figure 2C in the main text.

Figure S3: PP1 is displayed in white surface in all figures except S3. It would make sense to change it to white.

This has been updated as requested.

Figure S4: Kd of PP17-330:I63A/Y64A I3 27-68 (Bottom left) is 15.5 on the figure and 15.6 in the text (page 9).

Corrected.

Figure S5B: Labelling of H148 missing

Added.

Figure S5D: Mn atoms displayed in red instead of purple in all other figures and it might be a good idea to use a different color for SDS22 as green is the color of caspase in figure S3.

The metals are now illustrated in light orange throughout the figures.

Reviewer #4 (Remarks to the Author):

The manuscript by Srivastava and colleagues describes the structural and biophysical aspects of Protein Phosphatase 1 inhibition by Inhibitor-3. The mechanistic aspects of this interaction differ from other PP1 inhibitors, but more importantly show how different combinations of interaction motifs can combine to achieve biological function. The crux of the paper is the description of a novel CCC motif that inhibits the active site and forms a fuzzy interaction with metal ions. The presented research is interesting and merits a publication, if CCC motif indeed makes fuzzy interaction with metal ions. However In my opinion some aspects of the paper, particularly the fuzzy binding of CCC motifs should be supported by additional data, given that such a mode of interaction has not been observed before.

We thank reviewer 4 for her/his strong support, especially that the presented research is interesting.

Main issues:

I have some reservations regarding the conclusion that CCC makes fuzzy interactions with metal ions, although it seems that several experiments indeed point in this direction. However, since the authors suggest that this is one of the main findings of the paper, the case should really be solid. Here are a few comments regarding the CCC motif:

-main point: after establishing that I3 and I3(27-68) have identical affinity and activity most of the experiments are made on the I3(27-68) construct. In particular, the mutations C>S dissecting the role of CCC motif are also made in truncated version of I3 (I327-68). In this construct the CCC motif is near the C-end of polypeptide, but in the full-length I3 the motif is in the center of the chain. I think it is necessary to confirm some of the mutagenesis results on the full-length I3. This is because higher dynamics/fuzziness of CCC motifs could also be the consequence of the motif being placed at the chain end, since it is known that chain termini tend to have increased flexibility. The data on figure 1e show that residues beyond 68 have reduced intensity, thus they may interact with the PP1 and make the CCC motif less flexible. The bottom line is that some of the mutants in Figure 3 need to be verified in the context of full-length I3, to ensure that fuzziness is not a consequence of increased dynamics due to placing the CCC motif to the C-terminus.

As requested, we created a I3 1-126 CCC→SSS variant and repeated the SPR measurements. The measurement showed that the SSS variant functions in a highly similar manner in both full-

length I3 and the minimal functional I3 domain (residues 27-68). For the I3 minimal binding construct 27-68, the K_D changes from ~10 nM to ~400 nM for the SSS variant, while for I3 1-126 the K_D changes from ~15 nM to ~300 nM; the data are highly consistent. The overall somewhat stronger binding of I3 1-126 SSS is likely due to dynamic charge:charge interactions driven by charged I3 residues ~70-100. This interaction itself is very weak and transient and can only be detected by NMR spectroscopy. Indeed, we tested if an I3 peptide including only residues 79-91 (includes most charged residues) can bind PP1, but we were unable to detect the interaction using SPR (see Figure R2 herein). Thus, this interaction only occurs when other I3 SLiMs, especially the RVxF motif, are present and able to bind PP1. These data have been added to the revised manuscript.

-second point: NMR experiments were performed at low pH (pH<7), SPR binding experiments at high pH (pH>7). (For ITC I cannot find in which buffer they were made, see comment below). The pKa of cysteine is in a range of 7-8, which suggests that cysteine may have had different protonation state in these experiments. Is there a particular reason for using high pH in binding experiments? How does the protonation state affect interactions of CCC motif? It is well known that reactivity of cysteine dependent on pH, due to liked equilibrium. I suggest that authors verify how pH affects the CCC motif interaction as this might importantly enhance the understanding of nature of this interaction.

As requested, we performed I3:PP1 SPR at varying pH – 6.5, 7.0, 7.5 and 8.0. The data at pH 7.5 and 8.0 shows identical binding – the pH 6.5 and 7.0 data is slightly (2-fold) weaker binding but still exceedingly strong (10 to 20 nM) – but this is due to technical reasons – we bind the proteins to the SPR sensor surface using His₆-tag coupling. At pH 6.5 and 7.0 this coupling is weaker due to the unfavorable pKa of histidine. Thus, the pH does not (or only minimally) affect the protein:protein interaction.

It should be mentioned that the pH values for each experiment are chosen to get the most rigorous and reproducible data. In general, over the last years we have shown that the optimal pH for ITC/SPR of regulators with PP1 is pH 8.0. All additional pH dependent data have been added to Figure S2 and are also discussed in the manuscript.

-are there any NOEs for the residues near the CCC motif and PP1?

-PRE-NMR would be very useful to get a more detailed picture of the CCC-PP1 interactions, if metal ion is substituted to a paramagnetic ion (as in recombinant protein) this could be used to obtain a direct measurement of CCC-active site distances.

We agree with the suggestions of the reviewer. We have tried for many years (efforts by multiple graduate students and postdoctoral coworkers) to produce PP1 for NMR spectroscopy. However, the yields of PP1 are very low (1-2 mg/Liter of *E. coli* culture). We have attempted to produce (²H,¹⁵N)-labeled PP1, by expressing 4 liters (16 flasks with 250 ml of M9/D₂O medium) of deuterated culture and despite very careful cell adaptation to growth in D₂O, we were only able to get ~1.5 mg of deuterated labeled PP1, which was enough to record a reasonable 2D [¹H,¹⁵N] TROSY spectrum. We next attempted to change the active site metals to Zn²⁺ to get a diamagnetic

PP1 [¹H, ¹⁵N] TROSY spectrum, however the exchange is difficult, and the spectral quality was very low due to limited PP1 yields.

Mixing NMR-active I3 with PP1 leads to many missing I3 cross peaks in the I3 2D [¹H, ¹⁵N] HSQC spectrum, either due to the increased molecular weight of the complex (directly interacting I3 residues) or due to proximity of I3 residues due the active site, which has two paramagnetic Mn²⁺ ions bound.

It is for exactly this reason that we used all the orthogonal techniques to understand the interaction in as much detail as possible.

-Is CCC motif conserved in I3 homologs? Perhaps this is missing in the discussion, if the fuzzy interactions with CCC motifs and methal are something more general or a unique feature of I3

Yes, the CCC motif is conserved in I3 – some homologues only have a CC motif, i.e., in yeast (Cys60/Cys61). Importantly, deletion of the CC motif in yeast I3 (called YIP1) is lethal.

-there is very limited information provided on the structure of CCC motif based on the crystallographic data. The structure of the shorter I3-PP1 complex is described in the second paragraph of the results section. Interactions between all motifs are described in detail, but the structural details for the interactions between CCC motif and PP1 are practically omitted. What are the distances between cystelins and metal ions? Is the problem because the B-factors are very high in this region? Is it possible to conclude anything from the crystal structure regarding the interactions of CCC with PP1, if not why? On figure 4C, for example how far are thiol groups from metal? Can interaction be expected based on the distance?

We have increased the detail in the description of the CCC motif interactions of the crystal structure. We also now report the distances between the metal ions and the sulfur atom of C60 (7.4 Å and 8.2 Å) in Figure 2F. However, the high B factors strongly suggest that these residues are dynamic, adopting multiple conformations over the active site (Figure 4A). Notably, Lys59 has some of the lowest B-factors in I3. Thus, the position of these three cysteines, while they are themselves dynamic, are still 'locked' in position in front of the active site due to the highly ordered and structured Lys59. The extent to which the three cysteines move towards (or further away) from the active site is unknown from the crystal structure.

-page 6. Reduction of intensities was used to locate the regions of I3 that bind to PP1. Authors conclude that 2 regions in RVxF dead construct interact with PP1. However the data on figure 1E also shows a third region (around residue 80) with reduced intensities. Can you comment on that and explain why this region does not interact with PP1?

Our SPR data show that the binding K_D of I3 residues 27-68 with PP1 is identical to that of full-length I3. Furthermore, IC50 measurements show that I3 residues 27-68 are necessary and sufficient to inhibit PP1 in a manner identical to full length I3. Thus, 27-68 constitute the full, biologically relevant PP1 interaction domain of I3.

Figure R2: SPR sensorgrams for I3₇₉₋₉₁ (peptide) and PP1 α ₇₋₃₃₀

As noted by the reviewer, a third region around residue 80 also shows reduced intensities. The I3 residues around residue 80 are highly charged and likely weakly engage PP1 via a dynamic charge:charge (fuzzy) interaction (the surface of PP1 is highly charged). However, this interaction does not lead to a functional effect on PP1, i.e., it does not influence the PP1 binding or inhibition. Furthermore, as can be seen in Figure R2, this region fails to bind PP1 in the absence of the other I3 SLiM interactions (we used a high 10 μ M I3 peptide concentration in this experiment and still got no SPR response). This is now also discussed in the manuscript.

-design of 'dead' mutations. There is one point in the design and interpretation of mutant effects that is a bit misleading. This is the issue with deletions or mutations of motifs. I don't think that deletion of motifs (eg. CCC dead) and SSS mutant are directly comparable. This is because for IDPs it is often the case that chain entropy plays an important role in resulting affinity. To state simply interactions by amino acids make favourable interactions, but backbone gives unfavourable entropic contribution. When you mutate residues you remove interactions but retain backbone/entropic component. However in deletions you remove both, that is why deletions can have much less pronounced effects than mutations of motifs. This is relevant for your conclusion regarding the importance of SILK vs. RVxF motifs. The SILKdead variant has identical affinity compared to wt, which is likely because chain deletion was used in this case. On the other hand for RVxF motifs its functionality was inspected via KAEA mutation. If the functionality of these two motifs is compared that an appropriate mutation of SILK is needed.

In response to this suggestion and comments of reviewer 1, we have performed additional experiments in which the SILK motif is mutated within the context of the full protein. Namely we generated the I3 27-68 SLTIK \rightarrow SLAAA construct (mutating interacting residues instead of simply deleting these residues) and the same variant in full length I3. Then we repeated the SPR experiments and tested for binding with PP1. As discussed in the response to reviewer 1, the

experiments showed similar results as the SILK_{dead} (deletion) mutant regarding binding strength (K_D) if I3 with PP1. We have now included these new data into the revised manuscript and clarified the contribution of the I3 SILK motif. These data are included in Figure S2.

-ITC: what is the buffer used for ITC experiments?

The ITC buffer composition is now included in the methods section – thank you for highlighting this omission.

-ITC: it would make sense to make a control titration with SSS variant of some of the S/C variants to verify that cysteine indeed bind metals. The NMR data (Figure 4E) shows a reduction of intensities also for the residues before CCC motif (50-60). Could be Histidine 53 also be involved in binding? Histidine and cysteine are well known to coordinate metal ions.

As suggested, we performed an additional NMR experiment using a construct of I3 1-78 in which the CCC motif was mutated (CCC→SSS). As now shown in Figures 4E and 4F, no binding to Zn²⁺ is observed with this variant, as the peaks do not lose intensity upon Zn²⁺ titration. Thus, the CCC motif is strictly required for Zn²⁺ binding.

-ITC: please note that enthalpies in Table S3 should be considered as apparent, since cysteine-metal ion interaction likely involves exchange of protons, which cause significant enthalpy effect. This is why experiments are usually performed in buffer with low enthalpy of ionization.

Table S3 has been updated to reflect the suggestion of the reviewer.

Minor :

-Table 1: In my opinion Table 1 legend should contain also the following information: temperature (to which Kd values refer) and explanation of Fit and n columns.

Thank you – updated.

-Table 1: number of digits. Errors and the corresponding parameters are given with either 1, 2 or 3 significant digits. All errors should be rounded to either 1 or 2 significant digits uniformly and the corresponding parameters should be rounded accordingly.

Thank you – updated.

-Figure 1H: I found the description (FL) unnecessary and confusing in figure legend, since FL acronym is not really used in the text

Thank you – updated.

-Figure 1A caption. What is the meaning of the underlined sequence is not explained.

Thank you – updated.

-Figure 4 caption: 'Sensogram between ... and ... '. Sentence is incomplete?.

Thank you – updated.

-Figure 4 panel C : there is some text 'molar ratio' that does not belong here

Thank you – updated.

-Figure 4 panels F and G missing units for the Kd. Panel G missing x-axis label

Thank you – updated.

Taken together, we would like to thank all four reviewers for their careful and insightful comments, which greatly improved the manuscript. After fulfilling all requests from the reviewers and updating the manuscript accordingly, we hope that you will find this manuscript acceptable for publication in *Nature Communications*.

Sincerely,

Wolfgang Peti

References

1. Dancheck, B., Ragusa, M. J., Allaire, M., Nairn, A. C., Page, R., and Peti, W. (2011) Molecular investigations of the structure and function of the protein phosphatase 1-spinophilin-inhibitor 2 heterotrimeric complex. *Biochemistry*. **50**, 1238–1246
2. van den Boom, J., Kueck, A. F., Kravic, B., Müschenborn, H., Giesing, M., Pan, D., Kaschani, F., Kaiser, M., Musacchio, A., and Meyer, H. (2021) Targeted substrate loop insertion by VCP/p97 during PP1 complex disassembly. *Nat Struct Mol Biol*. **28**, 964–971
3. Salvi, F., Trebacz, M., Kokot, T., Hoermann, B., Rios, P., Barabas, O., and Köhn, M. (2018) Effects of stably incorporated iron on protein phosphatase-1 structure and activity. *FEBS Lett*. **592**, 4028–4038
4. Heroes, E., Rip, J., Beullens, M., Van Meervelt, L., De Gendt, S., and Bollen, M. (2015) Metals in the active site of native protein phosphatase-1. *J. Inorg. Biochem*. **149**, 1–5
5. Choy, M. S., Moon, T. M., Ravindran, R., Bray, J. A., Robinson, L. C., Archuleta, T. L., Shi, W., Peti, W., Tatchell, K., and Page, R. (2019) SDS22 selectively recognizes and traps metal-deficient inactive PP1. *Proc. Natl. Acad. Sci. U.S.A.* **116**, 20472–20481

REVIEWERS' COMMENTS

Reviewer #1 (Remarks to the Author):

In the revised manuscript the Authors addressed all my major and minor critical notes I had reviewing the original version. They made the necessary adjustments in the manuscript to all my concerns, therefore. I have no further criticisms and support acceptance of the manuscript for publication.

Reviewer #2 (Remarks to the Author):

The authors have made a thorough response to the inquiries raised by all four reviewers. Notably, there was consensus among the reviewers that the aspects of the manuscript most in need of additional experiments and/or rewriting pertained to the functional role of the CCC motif. The authors have done well to provide new data where it is possible and to contextualize why some of the envisioned experiments are not technically feasible. Based on my own expertise, I am in agreement with the authors regarding the limitations of NMR and x-ray crystallography as they relate to this system. I am satisfied that the biomedical significance of the hypothesized mechanism is sufficiently high and that the data available support the author's conclusions. Thus, I have no additional concerns regarding the publication of this manuscript.

Reviewer #3 (Remarks to the Author):

I carefully read the revised manuscript from Wolfgang Peti et al and they carefully addressed all my concerns, and they improved the figures.
I therefore recommend the manuscript for publication in Nature communication.

Reviewer #4 (Remarks to the Author):

The authors have adequately responded to all comments and adapted the manuscript accordingly. As such the paper merits publication as it reveals a novel type of fuzzy interaction between involving metal ions.